# High-Modality Multimodal Transformer: Quantifying Modality & Interaction Heterogeneity for High-Modality Representation Learning

**Paul Pu Liang[1], Yiwei Lyu[2], Xiang Fan[1], Jeffrey Tsaw[1], Yudong Liu[1], Shentong Mo[1], Dani Yogatama[3], Louis-Philippe Morency[1], Ruslan Salakhutdinov[1]**
[1]Carnegie Mellon University, [2]University of Michigan, [3]DeepMind

Reviewed on OpenReview: `https://openreview.net/forum?id=ttzypy3kT7`

## Abstract

Many real-world problems are inherently multimodal, from the communicative modalities humans use such as spoken language, gestures, and paralinguistics to the force, proprioception, and visual sensors ubiquitous on robots. While there has been an explosion of interest in multimodal representation learning, these methods are still largely focused on a small set of modalities, primarily in the language, vision, and audio space. In order to accelerate generalization towards diverse and understudied modalities, this paper studies efficient representation learning for *high-modality scenarios* involving a large set of diverse modalities. Since adding new models for every new modality or task becomes prohibitively expensive, a critical technical challenge is *heterogeneity quantification*: how can we measure which modalities encode *similar information* and *interactions* in order to permit parameter sharing with previous modalities? This paper proposes two new information theoretic metrics for heterogeneity quantification: (1) *modality heterogeneity* studies how similar 2 modalities $\{X_1, X_2\}$ are by measuring how much information can be transferred from $X_1$ to $X_2$, while (2) *interaction heterogeneity* studies how similarly pairs of modalities $\{X_1, X_2\}, \{X_3, X_4\}$ interact by measuring how much information can be transferred from fusing $\{X_1, X_2\}$ to $\{X_3, X_4\}$. We show the importance of these 2 proposed metrics in high-modality scenarios as a way to automatically prioritize the fusion of modalities that contain unique information or interactions. The result is a single model, HIGHMMT, that scales up to 10 modalities (text, image, audio, video, sensors, proprioception, speech, time-series, sets, and tables) and 15 tasks from 5 research areas. Not only does HIGHMMT outperform prior methods on the tradeoff between performance and efficiency, it also demonstrates a crucial scaling behavior: performance continues to improve with each modality added, and it transfers to entirely new modalities and tasks during fine-tuning. We release our code and benchmarks at: `https://github.com/pliang279/HighMMT`.

## 1 Introduction

Multimodal machine learning brings unique challenges for both computational and theoretical research given the heterogeneity of various data sources (Liang et al., 2022). While there have been impressive advances in modeling language, vision, and audio (Agrawal et al., 2017; Ramesh et al., 2021), advances in sensing technologies have resulted in many real-world platforms such as cellphones, smart devices, self-driving cars, healthcare technologies, and robots now integrating a much larger number of sensors such as time-series, proprioception, sets, tables, and high-frequency sensors (Frantzidis

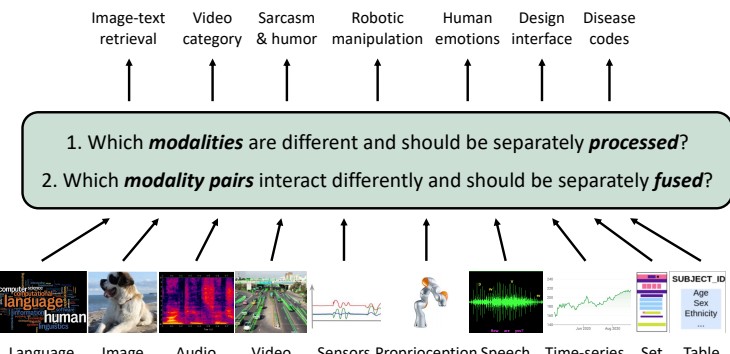

Figure 1: **Heterogeneity quantification**: Efficiently learning from many modalities requires measuring (1) *modality heterogeneity*: which modalities are different and should be separately processed, and (2) *interaction heterogeneity*: which modality pairs interact differently and should be separately fused. HIGHMMT uses these measurements to dynamically group parameters balancing performance and efficiency.

et al., 2010; Lee et al., 2019; Leiva et al., 2020; Liang et al., 2021a; Belpaeme et al., 2018; Yeong et al., 2021). This new setting of *high-modality learning* involves learning representations over many diverse modality inputs. As more modalities are introduced, adding new model parameters for every new modality or task (Tsai et al., 2019; Jayakumar et al., 2020; Lu et al., 2019) becomes prohibitively expensive and not scalable (Liang et al., 2022). A critical technical challenge for efficient high-modality learning, therefore, is *heterogeneity quantification*: how can we measure which modalities encode *similar information* and *similar interactions* in order to permit parameter sharing with previous modalities (see Figure 1)? For example, how can one determine whether the same modality encoder can be shared when processing language and speech, or that the same fusion network can be shared when fusing human speech and gestures as well as robot visual and force sensors?

In this paper, we propose a principled approach for heterogeneity quantification via modality information transfer, an approach that measures the amount of transferable information from one modality to another. Our first proposed metric, (1) *modality heterogeneity* studies how similar 2 modalities $\{X_1, X_2\}$ are by measuring how much usable information can be transferred from $X_1$ to $X_2$, and our second metric, (2) *interaction heterogeneity* studies how similarly 2 modality pairs $\{X_1, X_2\}, \{X_3, X_4\}$ interact by measuring how much usable interaction information can be transferred from $\{X_1, X_2\}$ to $\{X_3, X_4\}$. We show the importance of these 2 proposed metrics in high-modality scenarios as a way to automatically prioritize the fusion of modalities that contain unique information or unique interactions, and otherwise sharing parameters across similar modalities displaying similar information or interactions.

Operationalizing these ideas on a suite of 10 modalities, 15 prediction tasks, and 5 research areas, we show how to train a single model, HIGHMMT, that (1) improves the tradeoff between performance and efficiency over task-specific state-of-the-art models (Liang et al., 2021b; Jayakumar et al., 2020), and general multimodal models with full parameter sharing (Jaegle et al., 2021b; Hu and Singh, 2021; Akbari et al., 2021; Reed et al., 2022), (2) enables cross-modal transfer by pretraining on source tasks before transferring to new target modalities and tasks, and (3) is especially beneficial for low-resource scenarios (less training data and partially-observable modalities). Beyond these empirical results, we believe that our insights on quantifying heterogeneity and information sharing in multimodal models are independently useful for future work.

## 2 High-Modality Multimodal Transformer

In this section, we describe our overall approach for high-modality representation learning (see Figure 2). In §2.1, we formalize modality and interaction heterogeneity to understand whether modalities should be processed similarly or differently. Using these insights, §2.2 describes our proposed HIGHMMT model with dynamic parameter sharing based on heterogeneity measurements.

### 2.1 Measuring Heterogeneity via Modality Information Transfer

We begin our motivation by formalizing two important sources of heterogeneity in multimodal tasks. Firstly, *modality heterogeneity* occurs because the information present in different modalities often shows diverse qualities, structures, and representations. Secondly, *interaction heterogeneity* occurs because different modalities interact differently to give rise to new information when used for task inference. Formalizing and measuring these two sources of heterogeneity results in actionable insights for building multimodal models: measuring modality heterogeneity enables us to answer: should I use the same unimodal model to encode $X_1$ and $X_2$? Measuring interaction heterogeneity enables us to answer: should I use the same fusion model to fuse $\{X_1, X_2\}$ and $\{X_3, X_4\}$? We will formalize heterogeneity via *modality transfer*, an approach that measures the amount of transferable information from one modality to another.

**Estimating modality heterogeneity via unimodal information transfer.** We propose to measure heterogeneity between modalities $X_1$ and $X_2$ via unimodal transfer. Given a task $Y$ defined over $X_1$ and $X_2$, how well does an unimodal model trained on the task $(X_1; Y)$ transfer to $(X_2; Y)$? We choose model transfer as our focus of heterogeneity since it is captured at the level of features extracted via representation learning, rather than at the data-level. Even though the input data may be very different (e.g., images from different cameras or paraphrased sentences), effective feature extractors may be able to learn similar representations from them. Furthermore, it directly models task-relevance: the degree of heterogeneity depends on the end task, which enables using these heterogeneity measures subsequently for end-task optimization.

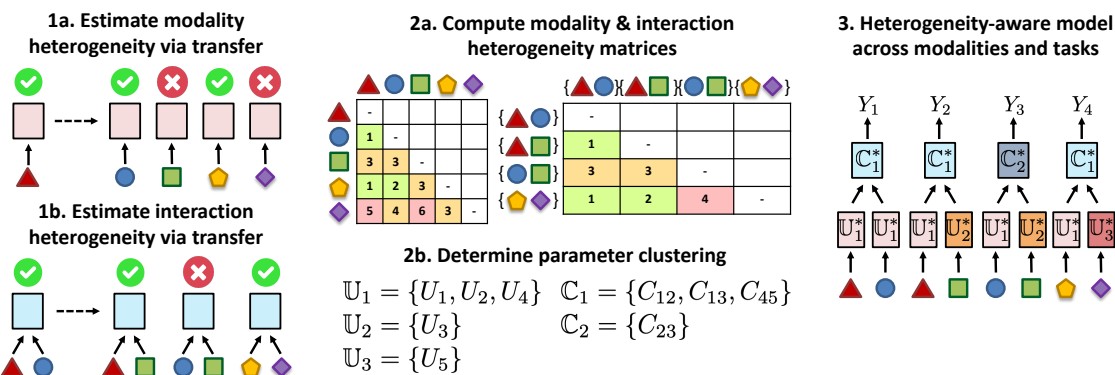

Figure 2: **HighMMT workflow**: (1) We estimate modality and interaction heterogeneity via modality transfer to determine which modalities should be processed and fused differently. (2) Using the inferred heterogeneity, we determine the optimal grouping of parameters balancing both total performance and parameter efficiency, which (3) informs our design of a heterogeneity-aware model with dynamic parameter sharing across many modalities and tasks. HIGHMMT enables statistical strength sharing, efficiency, and generalization to new modalities and tasks.

We formalize unimodal transfer as the difference in performance between unimodal models trained on $X_1$ before transfer to $X_2$, versus those trained directly on $X_2$. Specifically, we represent an unimodal model using modality $X_2$ with parameters $\theta$ as $\hat{y} = f(y|x_2; \theta)$. For a suitably chosen loss function $\ell(\hat{y}, y)$, define the loss of a model as $\mathbb{E}_{p(x_2, y)}\ell(f(y|x_2; \theta), y)$ which measures the expected error over the joint distribution $p(x_2, y)$. To measure transfer, we train 2 models to obtain an approximation of task performance: the first randomly initialized and trained on the target task giving loss $\mathcal{L}_2^*$,

$$\mathcal{L}_2^* = \min_\theta \mathbb{E}_{p(x_2, y)}\ell(f(y|x_2; \theta), y), \tag{1}$$

and the second using initialization from model parameters $\theta_1$ trained on the source task $(X_1; Y)$ before fine-training on the target task giving loss $\mathcal{L}_{1\to 2}^*$.

$$\theta_1 = \arg\min_\theta \mathbb{E}_{p(x_1, y)}\ell(f(y|x_1; \theta), y), \tag{2}$$

$$\mathcal{L}_{1\to 2}^* = \min_\theta \mathbb{E}_{p(x_2, y)}\ell(f(y|x_2; \theta \leftarrow \theta_1), y), \tag{3}$$

where $\theta \leftarrow \theta_1$ denotes parameter initialization with $\theta_1$. Intuitively, $\mathcal{L}_2^*$ measures the (baseline) task-relevant information in $X_2$, while $\mathcal{L}_{1\to 2}^*$ measures the task-relevant information transferrable from $X_1$ to $X_2$. The differences between these 2 losses,

$$T(X_1 \to X_2; Y) = \mathcal{L}_{1\to 2}^* - \mathcal{L}_2^*, \tag{4}$$

therefore measures the difficulty of transferring a model trained on the source task $(X_1; Y)$ to a target task $(X_2; Y)$. Note that computing $T(X_1 \to X_2; Y)$ only requires the training or fine-tuning of 2 models across the source and target modalities, which is efficient. In practice, the expectations over $p(x_1, y)$ and $p(x_2, y)$ are approximated using empirical samples from the training set (for model fine-tuning) and validation dataset (for final evaluation of performance).

What are some properties of $T(X_1 \to X_2; Y)$? For very different modalities $X_1$ and $X_2$, we typically expect a source task $(X_1, Y)$ to contain less usable information for a target task $(X_2; Y)$, which would imply that $\mathcal{L}_{1\to 2}^* \geq \mathcal{L}_2^*$ and therefore $T(X_1 \to X_2; Y) \geq 0$ (i.e., positive distance). This is consistent with work demonstrating negative transfer across different modalities (Liang et al., 2021b;c; Tulving and Watkins, 1974; Wang et al., 2019). Under these scenarios, the larger the positive magnitude of $T(X_1 \to X_2; Y)$, the more different modalities $X_1$ and $X_2$ are in the context of task $Y$ (more difficult to transfer). However, there can also be cases of zero or even positive transfer (i.e., $T(X_1 \to X_2; Y) \leq 0$), even in the surprising case of very different modalities (Lu et al., 2021). These cases reinforce the benefits of feature-based approaches to measure heterogeneity: while the raw modalities themselves seem very different, they can still be processed by similar models resulting in positive transfer, and should be assigned a difference of 0. Our final heterogeneity measure $d(X_1; X_2)$ aggregates the non-negative value (to account for positive transfer) of transfer difficulty statistics across both transfer directions $X_1 \to X_2$ and $X_2 \to X_1$:

$$d(X_1; X_2) = T(X_1 \to X_2; Y)_{\geq 0} + T(X_2 \to X_1; Y)_{\geq 0}. \tag{5}$$

where $x_{\geq 0} = \max(x, 0)$. Under certain assumptions on the modalities and tasks, our modality heterogeneity measure $d(X_1; X_2)$ is a metric: it satisfies *non-negativity*: $d(X_1; X_2) \geq 0$, with $d(X_1; X_2) = 0$ when $X_1 = X_2$, and *symmetry*: $d(X_1; X_2) = d(X_2; X_1)$, *positivity*, $X_1 \neq X_2$ implies that $d(X_1; X_2) > 0$, and a relaxed version of the *triangle inequality*: $d(X_1; X_3) \leq d(X_1; X_2) + d(X_2; X_3)$. However, in the most general case, there may be settings where positivity and the triangle inequality are not satisfied since the exact dynamics of transfer learning is still not well understood for general deep networks: positive transfer can happen (which would imply cases of $X_1 \neq X_2$ but $d(X_1; X_2) = 0$), and in practice, the relaxed triangle inequality is satisfied 96% of the time from a real heterogeneity matrix in Figure 5 (see Appendix A.1 for details).

**Estimating interaction heterogeneity via crossmodal information transfer.** We are also interested in interaction heterogeneity: specifically, how differently should I fuse modalities $\{X_1, X_2\}$ versus $\{X_3, X_4\}$? We therefore extend to crossmodal transfer by comparing the difference in performance between a multimodal model pretrained on $(X_1, X_2; Y)$ before transfer to $(X_3, X_4; Y)$, versus those trained directly on the target task $(X_3, X_4; Y)$. In other words, we measure the difference in loss between

$$\theta_{12} = \arg\min_{\theta} \mathbb{E}_{p(x_1, x_2, y)} \ell(f(y|x_1, x_2; \theta), y), \tag{6}$$

$$\mathcal{L}^*_{12 \to 34} = \min_{\theta} \mathbb{E}_{p(x_3, x_4, y)} \ell(f(y|x_3, x_4; \theta \leftarrow \theta_{12}), y), \tag{7}$$

and direct training

$$\mathcal{L}^*_{34} = \min_{\theta} \mathbb{E}_{p(x_3, x_4, y)} \ell(f(y|x_3, x_4; \theta), y), \tag{8}$$

to obtain $T(X_1, X_2 \to X_3, X_4; Y) = \mathcal{L}^*_{12 \to 34} - \mathcal{L}^*_{34}$. The distance $d(X_1, X_2; X_3, X_4)$ after aggregation over tasks and transfer directions estimates the interaction heterogeneity between $\{X_1, X_2\}$ and $\{X_3, X_4\}$.

**Modality and interaction heterogeneity matrix.** Finally, we construct a modality heterogeneity matrix $M_U(i, j) = d(X_i; X_j)$ and an interaction heterogeneity matrix (technically 4D-tensor) $M_C(i, j, k, \ell) = d(X_i, X_j; X_k, X_\ell)$. Determining parameter groupings to balance both total performance and parameter efficiency can be solved via agglomerative hierarchical clustering where modalities are nodes and heterogeneity measurements are edges. The number of clusters $k$ is treated as a hyperparameter dependent on the parameter budget (see Appendix A.3 for details, and see clustering examples in §3.1). Clustering on the modality heterogeneity matrix $M_U$ results in a grouping of modalities based on similarity (e.g., $\mathcal{U}_1 = \{X_1, X_2, X_4\}, \mathcal{U}_2 = \{X_3\}, \mathcal{U}_3 = \{X_5\}$), and likewise for the crossmodal matrix $M_C$ (e.g., $\mathcal{C}_1 = \{\{X_1, X_2\}, \{X_1, X_3\}, \{X_4, X_5\}\}, \mathcal{C}_2 = \{\{X_2, X_3\}, \mathcal{C}_3 = \{\{X_4, X_6\}, \{X_5, X_6\}\}$, and so on.

**Computational complexity.** In a high-modality setting, suppose we are given a suite of modalities and tasks of the form $\{(X_1, X_2, Y_1), (X_1, X_3, X_4, Y_2), ...\}$ and so on, where there are a total of $M$ unique modality and task pairs $\{(X_1, Y_1), (X_2, Y_1), (X_1, Y_2), (X_3, Y_2), (X_4, Y_2), ...\}$. In practice, the number of unique (pairwise) interaction and task pairs $\{(X_1, X_2, Y_1), (X_1, X_3, Y_2), (X_1, X_4, Y_2), (X_3, X_4, Y_2), ...\}$ is also $O(M)$, since the maximum number of modalities jointly observed for a task is never above a constant (at most 4 in all real-world datasets, and often 2 or 3). As an example in Figure 5, our experiments involve $M = 10$ modality and task pairs (across 4 tasks defined on $2, 2, 3$ and $3$ modalities respectively), and $8 = \binom{2}{2} + \binom{2}{2} + \binom{3}{2} + \binom{3}{2}$ interaction and task pairs.

The modality heterogeneity matrix for $M$ unique modality and task pairs has $M(M-1)/2$ unique entries after removing the upper triangular portion due to symmetry and diagonal entries since $d(X_i, X_i) = 0$. Computing these $M(M-1)/2$ entries exactly requires one to first train $M$ unimodal models (to estimate the $M$ $\mathcal{L}^*_m$ terms) before fine-tuning $M(M-1)$ transfer models (to estimate the $M(M-1)$ $\mathcal{L}^*_{m \to n}$ terms), for a total of $M^2$ pre-trained and fine-tuned models. The interaction heterogeneity matrix also requires $O(M^2)$ models for exact computation. However, we find that a key approximation can be made in practice: the heterogeneity matrices are highly structured due to distances approximately satisfying the triangle inequality, which implies that we do not need to compute all entries and instead rely on low-rank reconstruction from partial entries in practice. In our experiments, even using a low-rank approximation of $r = 3$ is sufficient to approximate the entire matrix. This suggests that we do not need to exhaustively measure unimodal and interaction transfer between all modality pairs to enjoy the benefits of our proposed approach. Instead, running a random sample of $O(M)$ pairs of heterogeneity values, and imputing the rest of the heterogeneity matrix, is sufficient in practice. Please see an example heterogeneity quantification for real-world datasets in §3.1, and Appendix A.2 for further details.

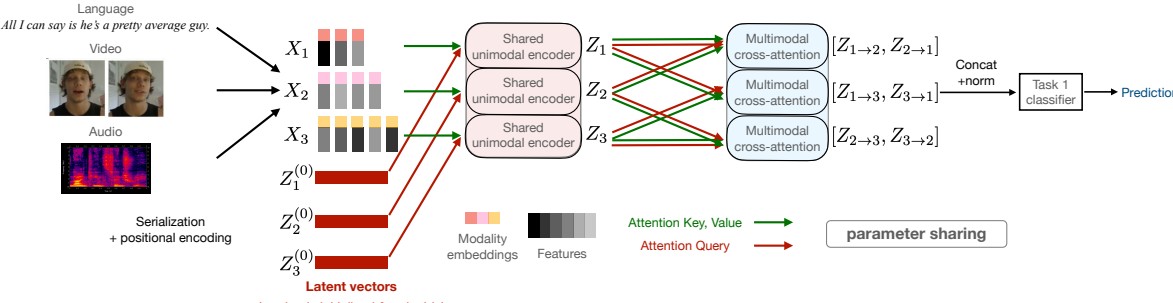

Figure 3: HιGHMMT architecture: Given arbitrary modalities, (1) the inputs are standardized into a sequence and padded, (2) modality embeddings and positional encodings are added to the input sequence, (3) a single shared unimodal Perceiver encoder is applied to all modalities to learn modality-agnostic representations, (4) each pair of unimodal representations is fed through a shared multimodal cross-attention layer to learn multimodal representations, and finally (5) all outputs are concatenated, batch-normalized, and fed into task-specific classification heads.

## 2.2 Capturing Heterogeneity and Homogeneity in HighMMT

Using these insights, we now describe our architecture for a general model HιGHMMT suitable for high-modality representation across many modalities and tasks (see Figure 3). Training the HιGHMMT model consists of 2 main steps (see Figure 4): (1) *homogeneous pre-training* of a fully shared model across all modalities, before (2) *heterogeneity-aware fine-tuning* to respect modality and interaction heterogeneity.

**Homogeneous pre-training**. We first design a homogeneous multimodal model fully shared across all modalities and tasks with the following key components (see Figure 3 and details in Appendix B).

*1. Standardized input sequence*: We first standardize modalities as a sequence of embeddings, as is already done for sequential data such as text, audio, and time series, and recently adapted for image patches too (Dosovitskiy et al., 2021). For tables, sets, and graphs we treat each element in the table/set/graph as an element in the sequence. The end result is a standardized input data $X_m$ of dimension $t_m \times d_m$, where $t_m$ is a modality and task-specific input sequence length, and $d_m$ is a modality and task-specific input dimension.

*2. Modality-specific embedding and positional encoding.* For each distinct modality $m \in M$ (which may appear across multiple tasks), we define a one-hot modality embedding $\mathbf{e}_m \in \mathbb{R}^{|M|}$, where $|M|$ is the total number of distinct modalities, to identify common modalities across different tasks for information sharing. We also introduce Fourier feature positional encodings $\mathbf{p}_m \in \mathbb{R}^{t_m \times d_{pm}}$, where $d_{pm}$ is the positional encoding dimension, to capture positional information across each modality. For multimodal tasks where a common dimension is shared across time (e.g., videos/time series), we apply a common positional encoding to capture the common time dimension (i.e., the first image frame occurs at the same time as the first word and first audio segment). Finally, the processed modality $m$ is given by concatenating $X_m = X_m \oplus \mathbf{e}_m \oplus \mathbf{p}_m \oplus \mathbf{0}_m$ (i.e., the input sequence, modality embedding, positional encodings and zero-padding) into a standard dimension $t_m \times d_{all}$. $d_{all} = \max_{m \in M}(d_m + |M| + d_{pm})$ where $d_m$ is the channel size of modality $m$, $d_{pm}$ is the positional encoding size of modality $m$, and $|M|$ is the modality encoding size (i.e., the total number of involved modalities).

*3. Shared unimodal networks.* Now that we have standardized all modalities into a common format, we design a general unimodal encoder with parameters $\mathbb{U}$ via a Transformer-based Perceiver block (Jaegle et al., 2021b). Our model recursively trains a latent array $Z_m$ of shape $d_{LN} \times d_{LS}$ (where $d_{LN}$ is the sequence length/number of latent vectors and $d_{LS}$ is the latent dimension) that is random initialized as $Z_m^{(0)}$. For each layer $L$ starting with a previously-computed representation $Z_m^{(L-1)}$, we first perform cross-attention from the processed input ($X_m$ of shape $t_m \times d_{all}$) to $Z_m^{(L-1)}$ obtaining an intermediate representation $\tilde{Z}_m^{(L)}$, before self-attention and feed-forward layers on $\tilde{Z}_m^{(L)}$ resulting in a new representation $Z_m^{(L)}$ for input to the next layer:

$$\tilde{Z}_m^{(L)} = \text{CROSS ATTENTION}(Z_m^{(L-1)}, X_m) = \text{softmax}\left(\frac{Q_c K_c^\top}{\sqrt{d_{LS}}}\right) V_c = \text{softmax}\left(\frac{Z_m^{(L-1)} W_{Q_c} W_{V_c}^\top X_m^\top}{\sqrt{d_{LS}}}\right) X_m W_{V_c},$$

$$\tag{9}$$

$$Z_m^{(L)} = \text{SELF ATTENTION}(\tilde{Z}_m^{(L)}) = \text{softmax}\left(\frac{Q_s K_s^\top}{\sqrt{d_{LS}}}\right) V_s = \text{softmax}\left(\frac{\tilde{Z}_m^{(L)} W_{Q_s} W_{V_s}^\top \tilde{Z}_m^{(L)\top}}{\sqrt{d_{LS}}}\right) \tilde{Z}_m^{(L)} W_{V_s},$$
$$\tag{10}$$

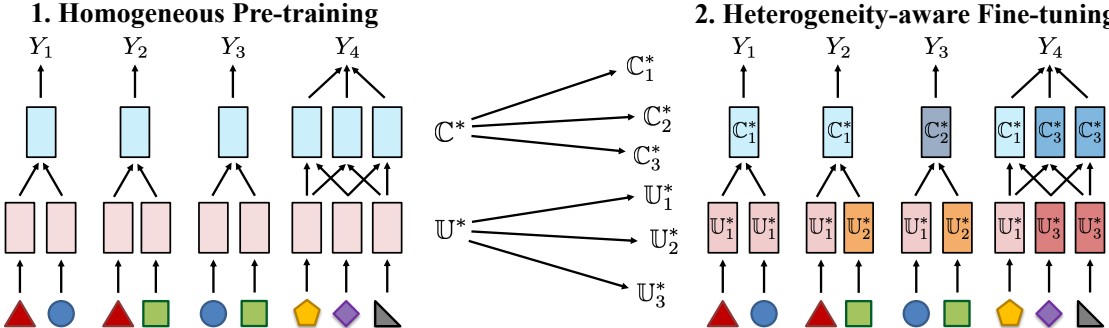

Figure 4: **HighMMT training** involves 2 steps: (1) *homogeneous pre-training* of a fully shared model across all modalities, before (2) *heterogeneity-aware fine-tuning* of modality and interaction parameters in different groups to respect modality and interaction heterogeneity respectively.

with trainable cross-attention parameters $W_{Q_c} \in \mathbb{R}^{d_{LS} \times d_{LS}}, W_{K_c} \in \mathbb{R}^{d_{all} \times d_{LS}}, W_{V_c} \in \mathbb{R}^{d_{all} \times d_{LS}}$ and self-attention parameters $W_{Q_s} \in \mathbb{R}^{d_{LS} \times d_{LS}}, W_{K_s} \in \mathbb{R}^{d_{LS} \times d_{LS}}, W_{V_s} \in \mathbb{R}^{d_{LS} \times d_{LS}}$. Repeating cross- and self-attention between the latent vector and the input modality summarizes the relationships between modality elements into the latent vector, resulting in a final unimodal representation $Z_m \in \mathbb{R}^{d_{LN} \times d_{LS}}$. Summarizing all information into a common $d_{LN} \times d_{LS}$ latent array regardless of the input shape $t_m \times d_{all}$ results in total runtime only linear with respect to the size of $t_m$ and $d_{all}$ which scales to high-modality scenarios.

*4. Shared crossmodal networks.* To learn multimodal representations, we use a shared Crossmodal Transformer block with parameters $\mathbb{C}$ (Tsai et al., 2019; Lu et al., 2019). Given 2 unimodal representations $Z_1$ and $Z_2$ of common shape $d_{LN} \times d_{LS}$ learned from unimodal Perceiver encoders, a Crossmodal Transformer (CT) block uses crossmodal self-attention by setting the input layer query $Q = Z_1$ and keys and values $K, V = Z_2$ to learn attention from $Z_2$ to $Z_1$, and a separate block to capture the attention in the opposite direction.

$$Z_{2 \to 1} = \text{CROSS ATTENTION}(Z_1, Z_2) = \text{softmax}\left(\frac{Q_1 K_2^\top}{\sqrt{d_k}}\right) V_2 = \text{softmax}\left(\frac{Z_1 W_{Q_1} W_{V_2}^\top Z_2^\top}{\sqrt{d_k}}\right) Z_2 W_{V_2}, \quad (11)$$

and vice-versa for $Z_{1 \to 2}$, with parameters $W_{Q_1}, W_{Q_2} \in \mathbb{R}^{d_{LS} \times d_k}, W_{K_1}, W_{K_2} \in \mathbb{R}^{d_{LS} \times d_k}, W_{V_1}, W_{V_2} \in \mathbb{R}^{d_{LS} \times d_k}$. This step enables one modality's elements to discover bidirectional interactions with another, resulting in a final multimodal representation $Z_{\text{mm}} = [Z_{1 \to 2}, Z_{2 \to 1}]$ of shape $d_{LS} \times 2d_k$. For each layer, we first perform cross-attention followed by self-attention and feed-forward functions. For tasks with more than 2 modalities, a Crossmodal Transformer block is applied for each pair of modalities before concatenating all representations.

*5. Task-specific classifier and multitask pre-training.* Finally, on top of $Z_{\text{mm}}$, we use a separate linear classification layer per task. To enable information sharing across modalities and tasks, homogeneous pre-training is performed across a diverse set of datasets in a multitask manner by optimizing a weighted sum of losses over tasks. The result is a single set of shared unimodal parameters $\mathbb{U}^*$ that encodes all modalities, and a single set of shared crossmodal parameters $\mathbb{C}^*$ that captures all pairwise interactions between modality pairs, along with all modality-specific embeddings $\mathbb{E}^*$ and task-specific classifiers $\mathbb{T}^*$.

**Heterogeneity-aware fine-tuning.** Finally, we account for heterogeneity by grouping unimodal parameters based on modalities that we know to be similar from §2.1 (e.g., setting $\mathbb{U}_1 = \{U_1, U_2\}, \mathbb{U}_2 = \{U_3\}, \mathbb{U}_3 = \{U_4, U_5, U_6\}$), and likewise for the crossmodal parameters (e.g., $\mathbb{C}_1 = \{C_{12}, C_{13}, C_{14}\}, \mathbb{C}_2 = \{C_{23}, C_{15}\}, \mathbb{C}_3 = \{C_{24}, ...\}$). From Figure 4, these parameter groups are first initialized with the homogeneous model $\mathbb{U}^*$ and $\mathbb{C}^*$ before separate fine-tuning, which results in final parameters $\mathbb{U}^* \to \{\mathbb{U}_1^*, \mathbb{U}_2^*, ...\}$ and $\mathbb{C}^* \to \{\mathbb{C}_1^*, \mathbb{C}_2^*, ...\}$. The modality embeddings $\mathbb{E}^*$ and task classifiers $\mathbb{T}^*$ are jointly fine-tuned as well. Fine-tuning is also performed in a multitask manner by optimizing a weighted sum of supervised losses across all modalities and tasks.

## 3 Experiments

**Setup**: In this section, we design experiments to analyze the multitask, transfer, and generalization capabilities of HIGHMMT. We use a large collection of multimodal datasets provided in MultiBench (Liang et al., 2021b) spanning 10 modalities, 15 prediction tasks, and 5 research areas. We trained 3 multitask models across

Table 1: We investigate a multitask setup to evaluate the performance of HIGHMMT across different modality inputs and prediction objectives. The total size of datasets involved in our experiments exceeds $370,000$ and covers diverse modalities, tasks, and research areas.

| Datasets | Modalities | Size | Prediction task | Research Area |
|---|---|---|---|---|
| ENRICO | {image, set} | $1,460$ | design interface | HCI |
| UR-FUNNY | {text, video, audio} | $16,514$ | humor | Affective Computing |
| MOSEI | {text, video, audio} | $22,777$ | sentiment, emotions | Affective Computing |
| MIMIC | {time-series, table} | $36,212$ | mortality, ICD-9 codes | Healthcare |
| PUSH | {image, force, proprioception, control} | $37,990$ | object pose | Robotics |
| AV-MNIST | {image, audio} | $70,000$ | digit | Multimedia |
| V&T | {image, force, proprioception, depth} | $147,000$ | contact, robot pose | Robotics |

**Modality heterogeneity matrix**

| | MIMIC – table | MIMIC – time series | AV-MNIST – image | AV-MNIST – audio | MOSEI – video | MOSEI – audio | MOSEI – text | UR-FUNNY – video | UR-FUNNY – audio | UR-FUNNY – text |
|---|---|---|---|---|---|---|---|---|---|---|
| MIMIC – table | - | | | | | | | | | |
| MIMIC – time series | 0.4 | - | | | | | | | | |
| AV-MNIST – image | 20 | 20 | - | | | | | | | |
| AV-MNIST – audio | 4.1 | 5.0 | 0.2 | - | | | | | | |
| MOSEI – video | 0 | 0.2 | 20 | 4.1 | - | | | | | |
| MOSEI – audio | 0 | 0 | 20 | 4.1 | 0 | - | | | | |
| MOSEI – text | 2.5 | 0.3 | 20 | 6.0 | 0.3 | 0 | - | | | |
| UR-FUNNY – video | 0 | 0.2 | 20 | 4.2 | 0 | 0 | 0.4 | - | | |
| UR-FUNNY – audio | 0.2 | 0.3 | 20 | 4.4 | 0 | 0 | 0.5 | 0.1 | - | |
| UR-FUNNY – text | 0 | 0.8 | 20 | 4.6 | 0 | 0 | 0 | 0 | 0 | - |

**Interaction heterogeneity matrix**

| | MIMIC – table + timeseries | AV-MNIST – image + audio | MOSEI – video + audio | MOSEI – video + text | MOSEI – audio + text | UR-FUNNY – video + audio | UR-FUNNY – video + text | UR-FUNNY – audio + text |
|---|---|---|---|---|---|---|---|---|
| MIMIC – table + timeseries | - | | | | | | | |
| AV-MNIST – image + audio | 23 | - | | | | | | |
| MOSEI – video + audio | 0.8 | 22 | - | | | | | |
| MOSEI – video + text | 0.4 | 24 | 0.2 | - | | | | |
| MOSEI – audio + text | 0.4 | 24 | 0.2 | 0.3 | - | | | |
| UR-FUNNY – video + audio | 0.4 | 22 | 0 | 0 | 0 | - | | |
| UR-FUNNY – video + text | 0.4 | 22 | 0 | 0 | 0.1 | 0 | - | |
| UR-FUNNY – audio + text | 0.5 | 22 | 0 | 0 | 0.2 | 0 | 0.9 | - |

Figure 5: Modality and interaction heterogeneity matrices color coded by distances, with green showing smaller distances and dark red larger distances. We find clear task outliers (AV-MNIST has high difficulty transferring to others), and that there is generally more interaction heterogeneity than unimodal heterogeneity. Otherwise, the same modality and modality pairs across different tasks are generally similar to each other.

combinations of these datasets (see Table 1 and Appendix C for details). Overall, the total size of datasets involved in our experiments exceeds $370,000$ and covers diverse modalities such as images, video, audio, text, time-series, robotics sensors, sets, and tables, prediction tasks spanning the image-caption matching, robot pose, object pose, robot contact, design interfaces, digits, humor, sentiment, emotions, mortality rate, and ICD-9 codes from the research areas of affective computing, healthcare, multimedia, robotics, and HCI.

## 3.1 Heterogeneity Measurements and Parameter Groups

We begin with a study of the heterogeneity matrices in Figure 5 and the resulting parameter groups.

**Modality heterogeneity:** We first notice that the modalities from AV-MNIST only transfer well to each other and has high difficulty transferring to other modalities from the other datasets. The same modality across different tasks is generally similar to each other (e.g., text between UR-FUNNY and MOSEI, audio between UR-FUNNY and MOSEI). The text modality in UR-FUNNY seems to be close to most other modalities, and likewise for the tabular modality in MIMIC. It is also worth noting that the video and audio modalities are not the most informative in MOSEI, and predictions are dominated by language (Zadeh et al., 2017), which may explain their general homogeneity with respect to other modalities.

**Interaction heterogeneity:** There is generally more interaction heterogeneity than unimodal, implying that the interactions between modality pairs tend to be more unique. Again, we notice the general poor transfer from the modality pair (image+audio) in AV-MNIST to other pairs, and the general strong transfer from (audio+text) in UR-FUNNY to the rest, which shows a relationship between modality and interaction heterogeneity. We also find that the same modality pairs (video+text) and (video+audio) shows crossmodal

similarity across both datasets they appear in: MOSEI and UR-FUNNY. Finally, while the triplet of crossmodal pairs in MOSEI are quite different from each other, those in UR-FUNNY are more similar.

Using these measurements, Appendix C.2 includes the final groups of parameters obtained after clustering the matrices for different values of $k$. As an example, for $|\mathbb{U}| = 3, |\mathbb{C}| = 3, k = 6$, the groups are $\mathcal{U}_1$ = {AV-MNIST image, AV-MNIST audio}, $\mathcal{U}_2$ = {MIMIC table, MOSEI video, MOSEI audio}, $\mathcal{U}_3$ = {MIMIC timeseries, MOSEI text, UR-FUNNY text, UR-FUNNY video, UR-FUNNY audio}, and $\mathcal{C}_1$ = {AV-MNIST image+audio}, $\mathcal{C}_2$ = {MOSEI video+audio}, and $\mathcal{C}_3$ = {MIMIC table+timeseries, MOSEI video+text, MOSEI audio+text, UR-FUNNY video+text, UR-FUNNY video+audio, UR-FUNNY audio+text}.

Finally, we observe the low-rank nature of the heterogeneity matrices due to symmetry and approximate triangle inequality, such that even using a low-rank approximation of $r = 3$ is sufficient to approximate the entire matrix. This suggests that we do not need to exhaustively measure unimodal and interaction transfer between all modality pairs to enjoy the benefits of our proposed approach (see more details in Appendix A.2).

### 3.2 Qualitative Results

We now present our results on the multitask, transfer, and generalization capabilities of HIGHMMT using performance and efficiency metrics. Henceforth, we will refer to the following models:

(1) **HighMMT share none** refers to individual copies of HIGHMMT models, one for each task.

(2) **HighMMT share all** refers to one single HIGHMMT model fully shared across all modalities and tasks.

(3) **HighMMT** refers to the full heterogeneity-aware HIGHMMT model across all modalities and tasks with learned parameter groupings based on heterogeneity measurements.

**Multitask performance and efficiency.** In Figure 6, we summarize the overall tradeoff between performance and efficiency using existing task-specific models and variants of HIGHMMT. The blue dots represent all possible combinations of task-specific models across multiple datasets (summarized in MultiBench (Liang et al., 2021b), $> 10^5$ total combinations) with their overall performance (scaled to a $0 - 1$ range before averaging across datasets) and overall efficiency (inverted total number of parameters). The red dots represent the state-of-the-art Pareto front: points that are not strictly dominated in both performance and efficiency. In light green, separate single-task HIGHMMT models (share none) already improve parameter efficiency as compared to standard Multimodal Transformers (Lu et al., 2019; Tsai et al., 2019). In dark green is HIGHMMT (share all) trained in a homogeneous multitask manner (i.e., with full parameter sharing across unimodal and multimodal layers within and across tasks), which further pushes forward the Pareto front by improving both performance and efficiency. Finally, in orange, HIGHMMT with heterogeneity-aware fine-tuning achieves significantly better tradeoffs between performance and efficiency, with efficiency and consistently high performance across multiple modalities and tasks.

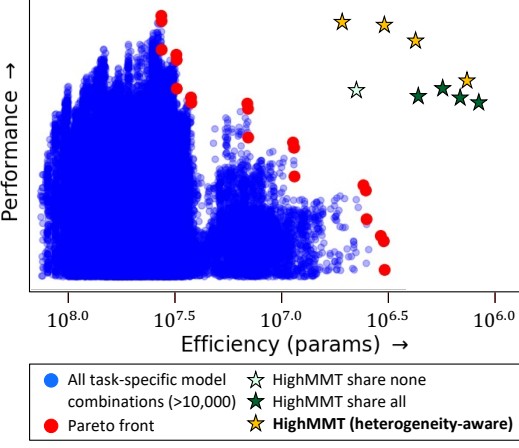

Figure 6: **Overall tradeoff.** HIGHMMT pushes forward the Pareto front of performance and efficiency as compared to all possible ($> 10^5$) combinations of task-specific models across multiple datasets (Liang et al., 2021b). The $x$-axis denotes (inverted) total parameters and $y$-axis denotes performance scaled to a $0 - 1$ range before averaging across datasets.

The suite of HIGHMMT models is obtained by tuning $k$, the total number of unimodal and crossmodal parameter groups (i.e., the number of clusters when clustering heterogeneity matrices). $k$ can be seen as a hyper-parameter depending on the computational budget, with smaller $k$ implying more parameter sharing on lower budgets and vice-versa. In Table 2, we show the effect of $k$ on average performance and total parameters. We test $k$ in the range $\{2, 4, 6, 7, 9\}$, with $|\mathbb{U}| = 1, |\mathbb{C}| = 1$, $|\mathbb{U}| = 3, |\mathbb{C}| = 1$, $|\mathbb{U}| = 3, |\mathbb{C}| = 3$, $|\mathbb{U}| = 3, |\mathbb{C}| = 4$, and

Table 2: Tuning the number of parameter groups results in controlled tradeoffs between parameters and performance.

| Clusters | Performance ↑ | Params (M) ↓ |
|---|---|---|
| 2 (share all) | $68.4 \pm 0.4$ | 1.07 |
| 4 | $68.8 \pm 0.5$ | 1.24 |
| 6 | $70.1 \pm 0.2$ | 2.47 |
| 7 | $71.0 \pm 0.1$ | 3.11 |
| 9 | $71.2 \pm 0.2$ | 4.23 |

Table 4: HIGHMMT achieves strong performance on overall performance and efficiency (mean and deviation over 10 runs), sometimes even beating (shown in **bold**) the task-specific state-of-the-art, especially on the relatively understudied modalities (time-series, robotics sensors, and sets) from the robotics (PUSH, V&T) HCI (ENRICO), and healthcare (MIMIC) research areas, while using **10× fewer parameters** due to parameter sharing and multitask learning. SOTA captures the max performance and parameters of more than 20 task-specific multimodal models: [1] GRADBLEND (Wang et al., 2020), [2] LF-LSTM (Ding et al., 2022), [3] LF (Gadzicki et al., 2020), [4] MULT (Tsai et al., 2019), [5] MFAS (Pérez-Rúa et al., 2019), [6] MFM (Yang et al., 2022), and [7] LRTF (Zeng et al., 2022).

| Model | ENRICO ↑ | PUSH ↓ | V&T ↑ | UR-FUNNY ↑ | MOSEI ↑ | MIMIC ↑ | AV-MNIST ↑ | Params (M) ↓ |
|---|---|---|---|---|---|---|---|---|
| SOTA | $51.0 \pm 1.4$[1] | $0.290 \pm 0.1$[2] | $93.6 \pm 0.1$[3] | $66.7 \pm 0.3$[4] | $\mathbf{82.1 \pm 0.5}$[4] | $68.9 \pm 0.5$[3,6,7] | $\mathbf{72.8 \pm 0.2}$[5] | 32.3 |
| HIGHMMT | $\mathbf{52.7 \pm 0.6}$ | $\mathbf{0.277 \pm 0.1}$ | $\mathbf{96.3 \pm 0.2}$ | $66.2 \pm 0.4$ | $80.2 \pm 0.2$ | $68.2 \pm 0.3$ | $71.1 \pm 0.2$ | $\mathbf{3.01}$ |

$|\mathbb{U}| = 4, |\mathbb{C}| = 5$ respectively where $|\mathbb{U}|, |\mathbb{C}|$ denote the number of unimodal and crossmodal parameter groups. We see a controllable tradeoff: starting with a fully shared model and increasing the number of parameter groups, we also see steadily improving performance approaching task-specific state-of-the-art models. Overall, optimizing for performance results in a model as strong as current state-of-the-art models while using 8× fewer total parameters. Optimizing for efficiency results in a model that reaches within 96% of current state-of-the-art performance but using 30× fewer total parameters (mean and deviation over 10 runs).

**Positive transfer to new modalities and tasks.** HIGHMMT also offers opportunities to study whether we can *transfer* knowledge between completely different modalities and tasks. Starting with the collection of 4 datasets in the order MOSEI, AV-MNIST, MIMIC, and UR-FUNNY ranked by largest dataset size (total of datapoints and memory storage per datapoint), we pre-train a fully-shared HIGHMMT model on 1/2/3 of the 4 tasks before fine-tuning on the fourth task only (e.g., train on MOSEI and transfer to UR-FUNNY, on MOSEI+AV-MNIST then transfer to UR-FUNNY, and on MOSEI+AV-MNIST+MIMIC then transfer to UR-FUNNY,

Table 3: **Cross-modal few-shot transfer to new modalities and tasks.** We train multitask HIGHMMT on 1/2/3 datasets and find that it generalizes few-shot to new modalities and tasks on the 4th dataset, with improved performance over single-task training on the 4th dataset. Cross-modal transfer improves with more pretraining tasks and works best on the smallest target tasks (UR-FUNNY).

| # Source tasks | Target task | | | |
|---|---|---|---|---|
| | UR-FUNNY | MOSEI | MIMIC | AV-MNIST |
| 0 (no transfer) | $63.1 \pm 0.5$ | $79.0 \pm 0.5$ | $67.7 \pm 0.6$ | $70.3 \pm 0.4$ |
| 1 | $63.5 \pm 0.5$ | $79.2 \pm 0.3$ | $67.9 \pm 0.5$ | $70.5 \pm 0.4$ |
| 2 | $64.0 \pm 0.7$ | $79.3 \pm 0.5$ | $68.0 \pm 0.8$ | $70.5 \pm 0.4$ |
| 3 | $\mathbf{64.7 \pm 0.4}$ | $\mathbf{79.6 \pm 0.6}$ | $\mathbf{68.4 \pm 0.6}$ | $70.6 \pm 0.4$ |

and likewise for transfer to the other 3 datasets (see full pre-train and transfer combinations in Appendix D.1).

From Table 3, we found that on all four combinations of multitask pretraining and fine-tuning, weights learned from other multimodal tasks generalize well to new modalities and tasks, improving performance over single target-task training (mean and standard deviation over 10 runs). When we increase the number of pretraining datasets, we observe a consistent improvement in fine-tuned target task performance. There is an inverse correlation between target task size and performance improvement: the smallest dataset, UR-FUNNY, benefited the most (+2.4%) from transfer learning from 0 to 3 multitask datasets. This implies that our multimodal pretraining-fine-tuning paradigm is useful for low-resource target modalities and tasks.

Finally, we compare transfer learning performance across different levels of partial observability. While one would expect the transfer to MIMIC to be the hardest due to its modality set {time-series, table} being completely disjoint from the remaining 3 datasets, we still observe a +0.8% gain as compared to single-task training. Therefore, HIGHMMT can generalize to new modalities and tasks. Unsurprisingly, for datasets with more overlap (e.g., UR-FUNNY with complete overlap in {text, video, audio} with respect to pretraining), we find larger improvements using transfer learning over single-task models (+2.4%).

**Comparison with task-specific state-of-the-art.** In Table 4, we compare multitask performance and efficiency with task-specific state-of-the-art models. We achieve performance within the range of published models (and usually close to the individual task-specific state-of-the-art) in MultiBench, which tallies more than 20 recent multimodal models in each task's literature (Liang et al., 2021b). In fact, HIGHMMT even sets new state-of-the-art results on several datasets, especially on the relatively understudied modalities (time-series, force and proprioception sensors, and sets) from the robotics (PUSH, V&T) and HCI (ENRICO) research areas. On top of strong performance, the main benefit lies in using fewer total parameters as compared to separate task-specific models - more than 10× reduction. Since this reduction grows with the number of tasks, our approach is scalable to high-modality scenarios.

Table 5: We conduct in-depth **ablation studies** and find strong evidence for (1) having separate unimodal and interaction layers, (2) determining parameter sharing via feature transfer, and (3) homogeneous pre-training before heterogeneity-aware fine-tuning into parameter groups (mean and standard deviation over 10 runs).

| | Model | UR-FUNNY ↑ | MOSEI ↑ | MIMIC ↑ | AV-MNIST ↑ | Ave ↑ |
|---|---|---|---|---|---|---|
| Full model | HIGHMMT | **66.2 ± 0.4** | **80.2 ± 0.2** | **68.2 ± 0.3** | **71.1 ± 0.2** | **71.4 ± 0.3** |
| Architecture ablations | - w/o embeddings | 63.0 ± 1.2 | 79.0 ± 0.7 | 67.1 ± 1.2 | 70.3 ± 0.7 | 69.8 ± 0.3 |
| | - w/o unimodal | 57.9 ± 0.3 | 61.9 ± 2.1 | 63.0 ± 0.9 | 59.5 ± 1.4 | 60.6 ± 0.7 |
| | - w/o crossmodal (Reed et al., 2022) | 63.8 ± 1.0 | 79.5 ± 0.5 | **67.9 ± 0.4** | 70.4 ± 0.5 | 70.4 ± 0.5 |
| Param sharing ablations | - share none (Liang et al., 2021b) | 63.7 ± 0.7 | 79.4 ± 0.4 | 67.7 ± 0.7 | 70.4 ± 0.1 | 70.2 ± 0.3 |
| | - share unimodal (Reed et al., 2022) | 62.5 ± 1.3 | 79.0 ± 1.1 | 63.4 ± 1.4 | 70.1 ± 0.7 | 68.8 ± 0.8 |
| | - share crossmodal (Akbari et al., 2021) | 63.0 ± 1.1 | 79.5 ± 0.3 | 64.3 ± 0.3 | 70.1 ± 0.9 | 69.2 ± 0.3 |
| | - share all (Singh et al., 2021) | 63.1 ± 0.7 | 79.2 ± 0.3 | 63.7 ± 1.6 | 68.6 ± 0.6 | 68.7 ± 0.5 |
| | - random difference | 62.9 ± 0.9 | 79.5 ± 0.6 | 67.6 ± 0.3 | 70.4 ± 0.2 | 70.1 ± 0.3 |
| | - feature difference (Sun et al., 2016) | 64.0 ± 1.0 | 79.4 ± 0.3 | **67.9 ± 0.3** | 70.1 ± 0.4 | 70.4 ± 0.2 |
| Training ablations | - w/o homogeneous pretraining | 61.2 ± 0.1 | 78.5 ± 0.1 | 64.8 ± 0.1 | **71.1 ± 0.2** | 69.9 ± 0.1 |

**Partial-observability.** Observe HIGHMMT performance on partially-observable modality subsets (i.e., target task involving modalities not present in the other tasks): from Table 4, we find that the model performs well on the MIMIC dataset despite its modality set {time-series, table} being completely disjoint from the remaining 3 datasets - we obtain similar performance across both multitask and single-task models $(68.2 ± 0.3\%$ vs $68.9 ± 0.5\%)$. We find that HIGHMMT multitask also works on ENRICO dataset in HCI $(52.7 ± 0.6\%$ multitask vs $51.0 ± 1.4\%$ single-task) despite it having completely disjoint modality inputs.

**Multitask fusion and retrieval.** We perform multitask training over multimodal fusion in AV-MNIST and retrieval in CIFAR-ESC. While fusion emphasizes information integration, retrieval focuses on aligning corresponding elements expressed through different views of the data (Liang et al., 2022). Even across these vastly different prediction tasks, we find that multitask training (60.5% retrieval accuracy) improves upon single-task training (58.8%). Not only have the unimodal networks simultaneously processed different modalities, but the crossmodal network has captured correspondences useful for both fusion and retrieval.

### 3.3 Ablation Studies

In this subsection, we carefully ablate the model architectures, parameter sharing, and training decisions.

**Architectural ablations.** We first analyze each architectural component of HIGHMMT: (1) *w/o embeddings* removes the only modality-specific component in the model - the modality embeddings. We set embeddings for all modalities to be the same to test whether a modality-specific component is necessary to capture heterogeneity across input data sources, (2) *w/o unimodal* removes the unimodal encoder and directly applies the cross-attention layer, and *w/o crossmodal* replaces the crossmodal layer with a concatenation of unimodal features and a linear classification layer. The latter resembles the most direct multimodal extension of existing work in shared unimodal encoders like Perceiver (Jaegle et al., 2021b), MultiModel (Kaiser et al., 2017), ViT-BERT (Li et al., 2021) or PolyViT (Likhosherstov et al., 2022). From Table 5, removing any of the 3 components in HIGHMMT results in worse performance. The unimodal encoder is particularly important.

**Param sharing ablations.** We further ablate with respect to possible parameter sharing settings in HIGHMMT: (1) *share none* uses separate unimodal and multimodal layers reminiscent of typical single-task multimodal transformers (Tsai et al., 2019; Lu et al., 2019; Hendricks et al., 2021), (2-3) *share unimodal (crossmodal)* only shares the unimodal (crossmodal) layer during multitask training, (4) *share all* shares all parameters without accounting for possible heterogeneity (Reed et al., 2022), (5) *random difference* determines $k$ parameter groups randomly rather than via heterogeneity measurements, (6) *feature difference* uses feature-level divergences on jointly trained unimodal encoders (i.e., $\|U(X_1) - U(X_2)\|_2^2$) rather than transfer performance to measure heterogeneity as is commonly done in transfer learning and domain adaptation (Daumé III, 2007; Sun et al., 2016). From Table 5, our proposed heterogeneity-aware parameter grouping results in the best overall performance as compared to fully shared, fully separate, or parameter grouping informed by other heterogeneity measures such as random or feature distance.

**Training ablations.** Finally, we explore *w/o homogeneous pretraining*: directly learning a model with parameter groups as selected by our approach as opposed to performing homogeneous pre-training before fine-tuning them into parameter groups. From Table 5, we find that this ablation underperforms - training parameter groups from scratch overfits to smaller datasets which hurts overall performance.

### 3.4 Understanding homogeneity and heterogeneity in HighMMT

We now take a deeper empirical analysis to better understand HIGHMMT, through parameter overlap and interference experiments. We investigate other model properties and visualizations in Appendix D.4.

**Parameter overlap.** Starting with a trained multitask HIGHMMT, we use a gradient-based method (Han et al., 2020) to determine how much each parameter is involved in a specific task. For each task $T$ and parameter $\theta \in \Theta$ in multitask model $M_\Theta$, we compute the involvement $I_T(\theta) = \mathbb{E}_{(x,y)\in T}|\nabla_\theta M_\Theta(y|x)|$ where $M_\Theta(y|x)$ is the predicted probability of correct target $y$ by $M_\Theta$ given $x$ as input. In other words, this measures the absolute gradient with respect to $\theta$ when predicting $y$ given $x$ in task $T$. A higher absolute gradient implies "activated" neurons and vice-versa for gradients closer to 0. This enables us to compute the extent a parameter $\theta$ is involved for each task. The *number of tasks* a given parameter $\theta$ is involved in can then be approximated by thresholding and summing up $n(\theta) = \sum_T \left( \mathbb{1}\{I_T(\theta) > \epsilon \max(I_1(\theta), I_2(\theta), I_3(\theta), I_4(\theta)\}\right)$ which returns an integer from 1 to 4. We chose a threshold $\epsilon$ such that parameters are classified as active about half the time on average, which occurs at $\epsilon = 0.2$.

Since we are interested in the level of parameter overlap in the shared unimodal encoder and multimodal layer, we set $\theta$ as these 2 modules and report results in Table 6. There is evidence of significant parameter overlap across unimodal encoders: more than 92% of neurons are involved in at least 3 of the 4 tasks. On the other hand, there is not nearly as much parameter overlap in the multimodal layer: only 10% of neurons are involved in 3 or 4 tasks. Hence, it seems like the unimodal encoders learn task-agnostic representations, but the subsequent multimodal layers (closer to task-specific classifiers) capture more task-specific information. This also reinforces our observation in §3.1 that there is generally more interaction heterogeneity than modality heterogeneity, which suggests using fewer unimodal parameter groups and more crossmodal parameter groups.

Table 6: We find evidence of significant **parameter overlap** across unimodal encoders: > 92% of neurons are involved in at least 3 of the 4 tasks, while the multimodal layers are more task-specific: only 10% of neurons are involved in 3 or 4 tasks.

| Component | Number of involved tasks | | | |
|---|---|---|---|---|
| | 1 | 2 | 3 | 4 |
| Unimodal layers | 2.8% | 5.1% | **61.1**% | **31.1**% |
| Crossmodal layers | **48.8**% | **39.7**% | 9.9% | 1.6% |

**Parameter interference.** Another empirical proof for parameter sharing in multitask models is the phenomenon of *parameter interference*: to what extent do parameters interfere with each other across tasks? We perform an experiment to investigate parameter interference: we pick one task and flip the labels in its training set, train the multitask model on the modified training set, and see how the incorrectly labeled task affects performance on other tasks. This experiment provides evidence of information sharing: if the multitask model does not share information (i.e., the model learns independent subspaces for each task), then one would not observe negative interference from one noisy dataset. We study negative interference under 3 configurations of training (a) the whole model; (b) only the unimodal encoder, and (c) only the multimodal layer on the flipped training set.

From Table 7, certain tasks are more affected by negative interference (e.g., AV-MNIST), while some tasks are not influenced as much (e.g., UR-FUNNY). Again, this reflects our heterogeneity measurements in §3.1, where AV-MNIST displays high heterogeneity. Furthermore, performance drops due to training the unimodal encoders are the most significant, which corroborates with our

Table 7: **Parameter interference**: we observe different performance drops on each task (columns) after training on one task with flipped labels (rows). Training the shared unimodal encoders causes the most harm, which implies that unimodal encoders contain more shared neurons sensitive to task changes. Red for drops greater than 20%, yellow for drops between 10 and 20%, and green for drops below 10%.

| (a) Training entire model | | | | |
|---|---|---|---|---|
| Flipped task | UR-FUNNY | MOSEI | MIMIC | AV-MNIST |
| UR-FUNNY | −24.6 | −8.83 | −10.6 | −57.7 |
| MOSEI | −4.07 | −59.7 | −20.3 | −53.2 |
| MIMIC | −3.59 | −5.83 | −33.1 | −37.5 |
| AV-MNIST | −3.50 | −1.23 | −4.87 | −68.9 |

| (b) Only training unimodal encoder | | | | |
|---|---|---|---|---|
| Flipped task | UR-FUNNY | MOSEI | MIMIC | AV-MNIST |
| UR-FUNNY | −23.8 | −10.1 | −12.8 | −58.4 |
| MOSEI | −5.77 | −57.6 | −21.1 | −52.7 |
| MIMIC | −3.03 | −3.54 | −35.0 | −56.3 |
| AV-MNIST | −2.94 | −7.82 | −53.6 | −69.3 |

| (c) Only training multimodal layer | | | | |
|---|---|---|---|---|
| Flipped task | UR-FUNNY | MOSEI | MIMIC | AV-MNIST |
| UR-FUNNY | −25.2 | −8.34 | −2.67 | −8.16 |
| MOSEI | 0.47 | −59.6 | −19.8 | −8.19 |
| MIMIC | 0.19 | −0.76 | −35.2 | −4.87 |
| AV-MNIST | −1.61 | −1.48 | −2.23 | −69.1 |

parameter overlap and heterogeneity analysis that unimodal encoders contain more entangled parameters which are more sensitive to task changes. On the other hand, multimodal layers contain more disentangled parameters, which results in higher heterogeneity measurements and needs more separate parameter groups.

## 4 Related Work

**Multimodal Transformers** have emerged as strong models for representation learning. Building upon the Transformer (Vaswani et al., 2017), multimodal extensions use either full self-attention over modalities concatenated across the sequence dimension (Li et al., 2019; Sun et al., 2019; Su et al., 2020; Chen et al., 2020) or a cross-modal attention layer (Lu et al., 2019; Tsai et al., 2019; Tan and Bansal, 2019), and are useful for sequential data by automatically aligning and capturing complementary features at different time-steps (Tsai et al., 2019; Yao and Wan, 2020; Lee et al., 2020c). Self-supervised multimodal pretraining has emerged as an effective way to train these architectures, with the aim of learning representations from large-scale unlabeled multimodal data before transferring to downstream tasks via fine-tuning (Lu et al., 2019; Li et al., 2019; Su et al., 2020). These pretraining objectives typically consist of unimodal masked prediction, crossmodal masked prediction, and multimodal alignment prediction (Hendricks et al., 2021).

**Unified encoder for unimodal learning.** Several works such as Perceiver (Jaegle et al., 2021a;b), MultiModel (Kaiser et al., 2017), ViT-BERT (Li et al., 2021), and PolyViT (Likhosherstov et al., 2022) have explored the possibility of using the same architecture for different inputs on unimodal tasks (i.e., language, image, video, or audio-only). The Transformer architecture has emerged as a popular choice due to its suitability for serialized inputs such as text (Devlin et al., 2019), images (Dosovitskiy et al., 2021), video (Sun et al., 2019), and time-series data (Lim et al., 2021), a phenomenon further observed by Lu et al. (2021) where a single Transformer pretrained on text transfers to sequence modeling and image classification. While these serve as building blocks in our model, our focus is on a general-purpose multimodal model for multitask and transfer learning across different subsets of modalities rather than unimodal tasks.

**Multimodal multitask and transfer learning.** There have also been several attempts to build a single model that works well on a suite of multimodal tasks (Li et al., 2019; Lu et al., 2019; Su et al., 2020; Cho et al., 2021; Reed et al., 2022). For example, UniT (Hu and Singh, 2021), VLBERT (Su et al., 2020), ViLBERT (Lu et al., 2019), and VL-T5 (Cho et al., 2021) are all unifying models for vision-and-language tasks. VATT (Akbari et al., 2021) jointly trains a shared model on video, audio, and text data to perform audio-only, video-only, and image-text retrieval tasks. FLAVA (Singh et al., 2021) found that pretraining a shared model with unpaired images, unpaired text, and image-text pairs results in strong performance on image-only, text-only, and image-text multimodal tasks, while Reed et al. (2022) scales up a single Transformer model for image, text, and decision-making tasks. However, all of these train a single model for all tasks, without investigating how heterogeneity can necessitate partial parameter sharing. On the transfer side, while more research has focused on transfer within the same modality with external information (Socher et al., 2013; Dunnmon et al., 2019; Xing et al., 2019; Zadeh et al., 2020), Liang et al. (2021c) is the only work that studies transfer to completely new modalities. However, they require paired data collection and modality-specific modeling. Our work goes beyond the commonly studied language, vision, and audio modalities to relatively understudied ones (e.g., tabular data, time-series, sensors, graphs, and set data). Furthermore, we show the possibility of generalizing to new modality subsets. Finally, our work also complements studies of transfer learning in a single modality (Standley et al., 2020; Wu et al., 2022; Zamir et al., 2018), where insights from task heterogeneity have informed multitask approaches, as well as multisensor fusion in various domains such as healthcare (Muhammad et al., 2021) and robotics (Suzuki and Matsuo, 2022; Taniguchi et al., 2023).

## 5 Conclusion

We propose an information transfer approach for estimating modality and interaction heterogeneity, a key component towards automatically determining which modalities should be processed and fused jointly for efficient representation learning in high-modality scenarios. Our resulting model, HIGHMMT dynamically determines the optimal parameter groupings balancing total performance and parameter efficiency, simultaneously achieves strong results on modalities (text, image, video, audio, time-series, sensors, tables, and sets) and tasks from different research areas, and transfers to new modalities and tasks during fine-tuning. We release our code and benchmarks which we hope will present a unified platform for subsequent analysis.

# Acknowledgements

This material is based upon work partially supported by Meta, National Science Foundation awards 1722822 and 1750439, and National Institutes of Health awards R01MH125740, R01MH132225, R01MH096951 and R21MH130767. PPL is partially supported by a Facebook PhD Fellowship and a Carnegie Mellon University's Center for Machine Learning and Health Fellowship. RS is supported in part by ONR N000141812861, ONR N000142312368 and DARPA/AFRL FA87502321015. Any opinions, findings, conclusions, or recommendations expressed in this material are those of the author(s) and do not necessarily reflect the views of the NSF, NIH, Meta, Carnegie Mellon University's Center for Machine Learning and Health, ONR, DARPA, or AFRL, and no official endorsement should be inferred. We are extremely grateful to the action editor and reviewers for their helpful discussions and feedback. Finally, we would also like to acknowledge NVIDIA's GPU support.

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

# Appendix

## A  Measuring Heterogeneity via Modality Information Transfer

### A.1  Modality and interaction heterogeneity measures

We formalize unimodal transfer as the difference in performance between unimodal models trained on $X_1$ before transfer to $X_2$, versus those trained directly on $X_2$. We represent an unimodal model using $X_2$ with parameters $\theta$ as $\hat{y} = f(y|x_2; \theta)$ and define the loss of a model as $\mathbb{E}_{p(x_2,y)}\ell(f(y|x_2; \theta), y)$ which measures the expected error over the joint distribution $p(x_2, y)$ under a suitably chosen loss function $\ell(\hat{y}, y)$. To measure transfer, we train 2 models: the first randomly initialized and trained on the target task giving loss $\mathcal{L}_2^*$,

$$\mathcal{L}_2^* = \min_\theta \mathbb{E}_{p(x_2,y)}\ell(f(y|x_2; \theta), y), \tag{12}$$

and the second using initialization from model parameters $\theta_1$ trained on the source task $(X_1; Y)$ before fine-training on the target task giving loss $\mathcal{L}_{1 \to 2}^*$.

$$\theta_1 = \arg\min_\theta \mathbb{E}_{p(x_1,y)}\ell(f(y|x_1; \theta), y), \tag{13}$$

$$\mathcal{L}_{1 \to 2}^* = \min_\theta \mathbb{E}_{p(x_2,y)}\ell(f(y|x_2; \theta \leftarrow \theta_1), y), \tag{14}$$

where $\theta \leftarrow \theta_1$ denotes parameter initialization with $\theta_1$. The differences between these 2 losses,

$$T(X_1 \to X_2; Y) = \mathcal{L}_{1 \to 2}^* - \mathcal{L}_2^*, \tag{15}$$

therefore measures the difficulty of transferring a model trained on the source task $(X_1; Y)$ to a target task $(X_2; Y)$. Our final heterogeneity measure $d(X_1; X_2)$ aggregates the non-negative value (to account for positive transfer) of transfer difficulty across both transfer directions $X_1 \to X_2$ and $X_2 \to X_1$:

$$d(X_1; X_2) = T(X_1 \to X_2; Y)_{\geq 0} + T(X_2 \to X_1; Y)_{\geq 0}. \tag{16}$$

where $x_{\geq 0} = \max(x, 0)$. We show that under certain assumptions on the modalities and tasks, our modality heterogeneity measure $d(X_1; X_2)$ is a metric:

**Non-negativity:** For any $X_1, X_2$,

$$d(X_1; X_2) = T(X_1 \to X_2; Y)_{\geq 0} + T(X_2 \to X_1; Y)_{\geq 0} \geq 0, \tag{17}$$

and when $X_1 = X_2$,

$$d(X_1; X_2) = T(X_1 \to X_2; Y)_{\geq 0} + T(X_2 \to X_1; Y)_{\geq 0} = 0 + 0 = 0. \tag{18}$$

**Symmetry:** For any $X_1, X_2$,

$$d(X_1; X_2) = T(X_1 \to X_2; Y)_{\geq 0} + T(X_2 \to X_1; Y)_{\geq 0} \tag{19}$$

$$= T(X_2 \to X_1; Y)_{\geq 0} + T(X_1 \to X_2; Y)_{\geq 0} \tag{20}$$

$$= d(X_2; X_1) \tag{21}$$

**Positivity:** In general, this heterogeneity measure may not satisfy positivity. In cases of positive transfer where using very different modalities as pretraining data can still help downstream tasks, we have that $\mathcal{L}_{1 \to 2}^* \leq \mathcal{L}_2^*$ (or $\mathcal{L}_{2 \to 1}^* \leq \mathcal{L}_1^*$) and as a result $T(X_1 \to X_2; Y) < 0$ (or $T(X_2 \to X_1; Y) < 0$) and $d(X_1; X_2) = 0$ even when $X_1 \neq X_2$. This distance metric satisfies positivity if we assume that only negative transfer exists:

**Assumption 1.** *(Negative transfer) For all $i, j$, we have that $\mathcal{L}_{i \to j}^* \geq \mathcal{L}_j^*$. In other words, for $i \neq j$, the loss on a task is always the lowest when directly trained a model for that modality and task, rather than pretraining on a different modality. When $i = j$, we have that $\mathcal{L}_{i \to i}^* = \mathcal{L}_j^*$ with equality.*

Under Assumption 1, for all $X_1 \neq X_2$, $\mathcal{L}_{1 \to 2}^* > \mathcal{L}_2^*$ and $\mathcal{L}_{2 \to 1}^* > \mathcal{L}_1^*$, so $d(X_1; X_2) = T(X_1 \to X_2; Y)_{\geq 0} + T(X_2 \to X_1; Y)_{\geq 0} > 0$, so positivity is satisfied. However, we would like to emphasize that even when positivity is not satisfied due to positive transfer, this is actually a benefit which tells us exactly when we are permitted to share parameters for different input modalities, since $d(X_1; X_2)$ is zero or small even though $X_1 \neq X_2$. From

the real heterogeneity matrices in Figure 5, we do find several examples of $X_1 \neq X_2$ where $d(X_1; X_2)$ is zero and this implies that we can safely share parameters across different modalities.

**Relaxed triangle inequality**: We will show a relaxed version of the triangle inequality (Sridharan and Kakade, 2008): for all $i, j, k$, there exists a positive constant $c \geq 1$ such that

$$d(X_i; X_k) \leq c(d(X_i; X_j) + d(X_j; X_k)). \tag{22}$$

To show the triangle inequality we will first need the following assumptions:

**Assumption 2.** *(Intermediate negative transfer) For all $i, j, k$, we have that $\mathcal{L}^*_{i \to j \to k} \geq \mathcal{L}^*_{i \to k}$. Intermediate negative transfer extends negative transfer: adding another pretraining task can only increase the loss.*

**Assumption 3.** *(Proxy task) For all $i, j, k$, there exists a positive constant $\lambda > 0$ such that*

$$\mathcal{L}^*_{i \to j} - \mathcal{L}^*_j \geq \lambda(\mathcal{L}^*_{i \to j \to k} - \mathcal{L}^*_{j \to k}). \tag{23}$$

*This assumption implies that we can use the differences in performance when transferred to a proxy task $k$ (i.e., $\mathcal{L}^*_{i \to j \to k} - \mathcal{L}^*_{j \to k}$) to estimate the differences in performance on the original source task $j$ (i.e., $\mathcal{L}^*_{i \to j} - \mathcal{L}^*_j$). Certainly these 2 quantities will not be exactly the same due to the different nature of modalities $j$ and $k$, but we assume that the relative differences are bounded below by constant $\lambda$.*

Under these assumptions, we will first show that the triangle inequality holds for both transfer directions, i.e.,

$$\mathcal{L}^*_{i \to k} - \mathcal{L}^*_k \leq c_1(\mathcal{L}^*_{i \to j} - \mathcal{L}^*_j + \mathcal{L}^*_{j \to k} - \mathcal{L}^*_k) \tag{24}$$

and

$$\mathcal{L}^*_{k \to i} - \mathcal{L}^*_i \leq c_2(\mathcal{L}^*_{k \to j} - \mathcal{L}^*_j + \mathcal{L}^*_{j \to i} - \mathcal{L}^*_i) \tag{25}$$

which would then imply

$$d(X_i; X_k) \tag{26}$$
$$= T(X_i \to X_k; Y)_{\geq 0} + T(X_k \to X_i; Y)_{\geq 0} \tag{27}$$
$$= \mathcal{L}^*_{i \to k} - \mathcal{L}^*_k + \mathcal{L}^*_{k \to i} - \mathcal{L}^*_i \qquad \text{(by neg. transfer)} \tag{28}$$
$$\leq c_1(\mathcal{L}^*_{i \to j} - \mathcal{L}^*_j + \mathcal{L}^*_{j \to k} - \mathcal{L}^*_k) + c_2(\mathcal{L}^*_{k \to j} - \mathcal{L}^*_j + \mathcal{L}^*_{j \to i} - \mathcal{L}^*_i) \tag{29}$$
$$\leq c(\mathcal{L}^*_{i \to j} - \mathcal{L}^*_j + \mathcal{L}^*_{j \to k} - \mathcal{L}^*_k + \mathcal{L}^*_{k \to j} - \mathcal{L}^*_j + \mathcal{L}^*_{j \to i} - \mathcal{L}^*_i) \qquad (c = \max(c_1, c_2)) \tag{30}$$
$$= c(T(X_i \to X_j; Y)_{\geq 0} + T(X_j \to X_i; Y)_{\geq 0} + T(X_j \to X_k; Y)_{\geq 0} + T(X_k \to X_j; Y)_{\geq 0}) \qquad \text{(by neg. transfer)} \tag{31}$$
$$= c(d(X_i; X_j) + d(X_j; X_k)) \tag{32}$$

To show each direction, we have that

$$\mathcal{L}^*_{i \to k} - \mathcal{L}^*_k \leq \mathcal{L}^*_{i \to j \to k} - \mathcal{L}^*_k \qquad \text{(by intermediate negative transfer)} \tag{33}$$
$$= \mathcal{L}^*_{i \to j \to k} - \mathcal{L}^*_{j \to k} + \mathcal{L}^*_{j \to k} - \mathcal{L}^*_k \tag{34}$$
$$\leq 1/\lambda(\mathcal{L}^*_{i \to j} - \mathcal{L}^*_j) + \mathcal{L}^*_{j \to k} - \mathcal{L}^*_k \qquad \text{(by proxy task)} \tag{35}$$
$$\leq c_1(\mathcal{L}^*_{i \to j} - \mathcal{L}^*_j + \mathcal{L}^*_{j \to k} - \mathcal{L}^*_k) \qquad (\text{for } c_1 = \max(1, 1/\lambda_1)) \tag{36}$$

and the other direction is symmetric. Since $c_1, c_2 = \max(1, 1/\lambda) \geq 1$ we have that the final constant $c = \max(c_1, c_2) \geq 1$.

In practice, the negative transfer and intermediate negative transfer assumptions may not hold. In fact, our experiments do show evidences of positive transfer and intermediate positive transfer: pretraining on very different modalities and tasks can help downstream modalities and tasks, and that this improvement grows are more pretraining modalities and tasks are used (see Table 3). Nevertheless, we find that the exact triangle inequality is approximately satisfied 87% (unimodal) and 86% (interaction) of the time from the heterogeneity matrices in Figure 5, and the relaxed version of the triangle inequality for constant $c = 5.0$ is satisfied 96% (unimodal) and 95% (interaction) of the time. This is also a contributing factor to the fact that the heterogeneity matrices have a low-rank structure that enables their approximation with only rank 3 matrices.

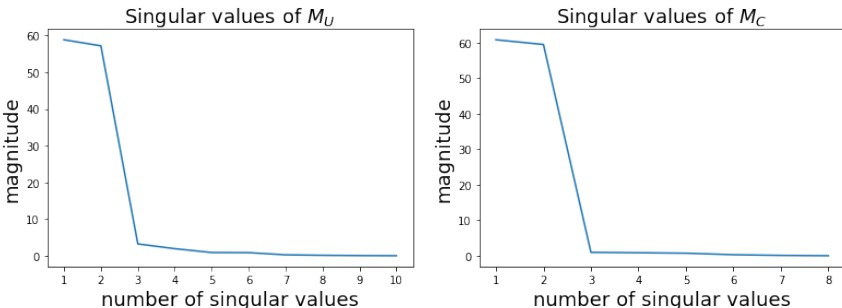

Figure 7: Distribution of singular values of the heterogeneity matrices $M_U$ and $M_C$. The top 3 singular values are sufficient to explain $\geq 97\%$ of the variance, so a low-rank approximation of $r = 3$ is sufficient to approximate the entire matrix. This suggests that simply testing a random set of $O(M)$ unimodal and interaction transfer values is sufficient to enjoy the benefits of our proposed approach.

## A.2 Modality and interaction heterogeneity matrix

In Figure 7, we show some experiments demonstrating the low-rank nature of the heterogeneity matrices $M_U$ and $M_C$. The top 3 singular values are sufficient to explain $\geq 97\%$ of the variance, such that even using a low-rank approximation of $r = 3$ is sufficient to approximate the entire matrix. For example, we can test a random set of $O(M)$ unimodal and interaction transfer values, and approximate the heterogeneity matrix $M = \sum_{i=1}^{h} u_i v_i^\top$ as an outer product of $k$ individual basis vectors $u_i$ and $v_i$, where $h$ is a smaller number than the actual dimension of $M$. This suggests that we do not need to exhaustively measure unimodal and interaction transfer between all modality pairs to enjoy the benefits of our proposed approach.

## A.3 Determining parameter groupings

We balance both total performance and parameter efficiency via agglomerative hierarchical clustering where modalities are nodes and heterogeneity measurements are edges. The number of clusters $k$ is treated as a hyperparameter dependent on the parameter budget. Clustering on the modality heterogeneity matrix $M_U$ results in a grouping of modalities based on similarity (e.g., $\mathcal{U}_1 = \{X_1, X_2, X_4\}, \mathcal{U}_2 = \{X_3\}, \mathcal{U}_3 = \{X_5\}$), and likewise for the interaction matrix $M_C$ (e.g., $\mathcal{C}_1 = \{\{X_1, X_2\}, \{X_1, X_3\}, \{X_4, X_5\}\}, \mathcal{C}_2 = \{\{X_2, X_3\}, \mathcal{C}_3 = \{\{X_4, X_6\}, \{X_5, X_6\}\}$, and so on. How can we choose the number of clusters $k$? Note that if $k$ is equal to the total number of modalities (or modality pairs) then it reduces to having separate models for each modality (and interaction), while $k = 1$ implies using a single model for all modalities and interactions. $k$ is therefore most suitably seen as a 'parameter budget' that one would like to control for efficiency. In our experiments, we explored a range of $k$ giving rise to a suite of models across controlled trade-offs between performance and efficiency (see Figure 6).

## B  HighMMT Details

At a high level, HIGHMMT includes the following components: (1) the inputs are standardized into a sequence and padded, (2) the perceiver input processing adds modality-specific modality embeddings and positional encodings to the serialized raw input; (3) the processed input from each modality is fed into a shared unimodal perceiver encoder; (4) each pair of unimodal perceiver output (unimodal representations) is fed through a shared crossmodal transformer layer twice (the first time with one modality as query and the other as context, and the second time vice versa); (5) finally, all outputs from multimodal layers are concatenated, batch-normalized to form a multimodal representation, and fed through a task-specific classification head to make a prediction. Figure 8 is an illustration of the high-level architecture.

### B.1 Perceiver Input Processing

We follow the data processing pipeline in the GitHub implementation for multimodal perceivers: `https://github.com/fac2003/perceiver-multi-modality-pytorch`. For each modality, we must specify in

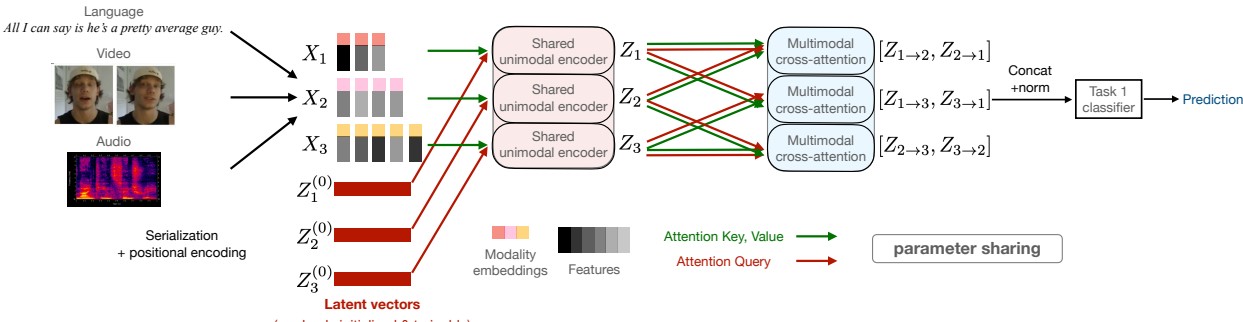

Figure 8: General architecture of HIGHMMT: Given arbitrary modalities, (1) the inputs are standardized into a sequence and padded, (2) modality embeddings and positional encodings are added to the serialized raw input, (3) a single shared unimodal Perceiver encoder is applied to all modalities to learn general-purpose representations regardless of the specific input modality, (4) each pair of unimodal representations is fed through a shared multimodal cross-attention layer twice (the first time with one modality as query and the other as context, and the second time vice versa) to learn general multimodal representations regardless of the input modalities and task, and finally (5) all outputs from cross-attention layers are concatenated, batch-normalized, and fed through a task-specific classification head to make a prediction. The unimodal encoders and multimodal layers are shared across tasks during multitask learning to enable statistical strength sharing, parameter efficiency, and quick generalization across diverse modalities and tasks.

advance the channel size (i.e., embedding size) and how many extra dimensions there are other than the channel/embedding dimension.

The modality embedding is just a one-hot vector denoting the index of the current modality, and the size of the vector is equal to the total number of modalities involved. This embedding layer identifies common modalities across different tasks to enable sharing of information. For example, the modality embedding of the image sequence for a video classification task will be shared with that of an input (static) image for an image and text question-answering task.

We also specify a few hyperparameters (such as num_freq_bands and max_freq) for the Fourier transformation used in the positional encoding. The positional encoding represents where this embedding is at through Fourier transformations (so if there is 1 extra dimension, then the positional encoding will encode the 1D position of each embedding; if there are 2 extra dimensions, then the positional encoding will encode the 2D position of each embedding). The positional encoding length can vary for each modality depending on the number of extra dimensions and the Fourier transformation hyperparameters.

The total embedding size of the processed output will be equal to $d_{all} = \max_{m \in M}(d_m + d_{pm} + |M|)$, where $M$ is the set of all modalities involved, $d_m$ is the channel size of modality $m$, $d_{pm}$ is the positional encoding size of modality $m$, and $|M|$ is the modality encoding size (i.e., the total number of involved modalities). When processing each modality, we concatenate the input channels, the positional encoding, and the modality encoding along the channel/embedding axis before adding zero-padding along this axis to match a desired total embedding size $d_{all}$. As a result, all modalities will be processed to have the same embedding size $d_{all}$. We also flatten all non-embedding dimensions so the processed input will always have shape $t_m \times d_{all}$ where $t_m$ is a modality-specific sequence length, and $d_{all}$ is the common embedding dimension.

For example, during multitask learning in the large setting (4 datasets involved: UR-FUNNY, MOSEI, MIMIC, and AV-MNIST), $d_{all} = 387$ (because the image modality from UR-FUNNY has a channel size of 371, positional encoding size of 7, and modality encoding size of 9). When processing the colorless image modality from AVMNIST ($7 \times 7 \times 16$), we have a channel size of 16, positional encoding size of 26, and modality encoding size of 9, so the processed output will be $49 \times 387$ where the first 16 dimensions along the last dimension represent 16 raw input dimensions, the next 336 dimensions are padded zeroes, the next 26 dimensions are positional encodings, and the final 9 dimensions are modality encodings.

Note that during this entire processing step all procedures are programmatic and there are no trainable parameters involved.

## B.2 Unimodal Perceiver Encoder

Now that we have standardized all modality inputs into a common representation, we follow the Perceiver architecture (Jaegle et al., 2021b) to perform modality and task-agnostic representation learning from each input modality. Starting with a latent array of shape $d_{LN} \times d_{LS}$ (array size configurable as a hyperparameter, where $d_{LN}$ is the number of latent vectors and $d_{LS}$ is the latent dimension) with trainable initialization, for each layer, we first perform cross-attention on the latent array using the processed input array (of shape $t_m \times d_{all}$) as context. Cross-attention between the latent vector and the input modality sequence learns relationships between elements in each modality, resulting in unimodal contextualized representations. The resulting latent array then goes through a latent transformer (with self-attention and feed-forward layers). We repeat this architecture for each layer within the encoder. The main advantage of this Perceiver encoder is that it can encode the input into a common $d_{LN} \times d_{LS}$ latent array regardless of the input shape $t_m \times d_{all}$, and the total runtime is linear with respect to the size of $t_m$ which scales to high-modality scenarios. Note that only one copy of a unimodal Transformer (Perceiver) block is used to encode all modalities simultaneously, which enables statistical strength sharing and general-purpose representation learning regardless of the specific input modality.

## B.3 Crossmodal Transformer layer

To learn modality and task-agnostic multimodal representations, we use multiple layers of a general-purpose Crossmodal Transformer block (Tsai et al., 2019; Lu et al., 2019). Given 2 unimodal representations $Z_1$ and $Z_2$ of common shape $d_{LN} \times d_{LS}$ learned from unimodal Perceiver encoders, a Crossmodal Transformer (CT) block uses crossmodal self-attention by setting the input layer query $Q = Z_1$ and keys and values $K, V = Z_2$ to learn attention from modality 1 to modality 2, and a separate block to capture the attention in the opposite direction. This step enables one modality's sequence elements to discover correspondences in another. A Crossmodal Transformer block using $Z_1$ to attend to $Z_2$ (and vice-versa) results in a multimodal representation $Z_{\mathrm{mm}} = [Z_{1 \to 2}, Z_{2 \to 1}] = [\mathrm{CT}(Z_1, Z_2), \mathrm{CT}(Z_2, Z_1)]$. For each layer, we first perform cross-attention followed by self-attention and feed-forward functions. In the end, we only take the last $d_{LS}$-dimensional vector out of the $d_{LN} \times d_{LS}$ final latent array as the output of this module. For tasks with more than 2 modalities, a Crossmodal Transformer block is applied for each pair of modalities before concatenating all multimodal representations. Again, only one copy of a multimodal layer is used on all tasks to learn general representations regardless of the input modalities and task.

## B.4 Task-specific classifiers

Since each task may have a different number of modalities and output classes, we create a separate classification head for each task. For each classification head, it concatenates all outputs of the Crossmodal Transformer layer (so 2-modality tasks have concatenated size of $2d_{LS}$, 3-modality tasks have concatenated size of $6d_{LS}$, etc), performs batch-normalization, and feeds the normalized multimodal representation $Z_{\mathrm{mm}}$ into a linear layer that maps to the logits for this task. This classification layer composes individual correspondences learned within and across modalities to form a final prediction.

## B.5 Homogeneous multitask pre-training

Since each task has a different number of training batches, not all tasks will be involved in each training step. We arrange the tasks to be included in each training step such that more tasks will be trained simultaneously towards the end of an epoch. For example, if task $A$ has 300 training batches, task $B$ has 200 training batches, and task $C$ has 100 training batches, then for the first 100 training steps in an epoch, only task $A$ will be used; then for the next 100 steps both $A$ and $B$ will be used; and for the last 100 steps, all three tasks will be used. This approach tends to work better than including all tasks in all steps via uniform batch sampling because the task with fewer training batches tends to overfit in the latter approach.

Within each training step, we compute the losses of the batch from each task used and compute the gradient using a weighted sum of the losses. The weights are part of the hyperparameters that we can tune to ensure balanced training. Then we update the model using the computed gradients.

We compute validation performance after each epoch for each task, and aggregate validation performances across all tasks (this is necessary because different tasks are measured differently, sometimes bigger is better, sometimes smaller is better). When all tasks are accuracy-based (such as the large setting), we just weigh them equally. Then we report test performance on the checkpoint with the highest aggregated validation performance.

The result from homogeneous multitask pre-training is a set of modality embeddings, common unimodal and crossmodal parameters $\mathbb{U}^*$ and $\mathbb{C}^*$, and individual task classifiers.

### B.6  Heterogeneity-aware fine-tuning

We account for heterogeneity by grouping unimodal parameters based on modalities that we know to be similar from §2.1 (e.g., setting $\mathbb{U}_1 = \{U_1, U_2\}, \mathbb{U}_2 = \{U_3\}, \mathbb{U}_3 = \{U_4, U_5, U_6\}$), and likewise for the crossmodal parameters (e.g., $\mathbb{C}_1 = \{C_{12}, C_{13}, C_{14}\}, \mathbb{C}_2 = \{C_{23}, C_{15}\}, \mathbb{C}_3 = \{C_{24}, ...\}$). These groups of parameters are first initialized with the homogeneous model $\mathbb{U}^*$ and $\mathbb{C}^*$ before separate fine-tuning, which results in final parameters $\mathbb{U}^* \to \{\mathbb{U}_1^*, \mathbb{U}_2^*, ...\}$ and $\mathbb{C}^* \to \{\mathbb{C}_1^*, \mathbb{C}_2^*, ...\}$. The modality embeddings and task classifiers are jointly fine-tuned as well.

### B.7  Transfer learning details

If we are trying to transfer from tasks $\{A, B, C\}$ to $D$, initially we start with a randomly initialized HIGHMMT model that defines modality embeddings for all modalities in $\{A, B, C, D\}$ as well as a classification head for each. Then, we pretrain the model using multitask learning on $\{A, B, C\}$ using the same procedure as before. After saving a good checkpoint as measured by aggregated validation performance on pretraining tasks $\{A, B, C\}$, we finetune the trained model on target task $D$. The modality and tasks in $\{A, B, C\}$ present during multitask pretraining can be very different from those encountered in $D$ during fine-tuning.

### B.8  Few-shot multitask learning details

We also investigated few-shot learning using limited labeled data in a target task. When we perform few-shot learning on task $D$ with the help of tasks $\{A, B, C\}$, we jointly train $\{A, B, C, D\}$ together in the same multitask manner as before, but since we don't care about the performance of our model on auxiliary tasks $\{A, B, C\}$, we assign a higher weight to the losses on task $D$ and keep track of the best validation performance on $D$ when selecting checkpoints.

## C  Experimental Setup

In this section, we provide additional details on the experimental setup to analyze the multitask, transfer, and generalization capabilities of HIGHMMT.

### C.1  Setup

We use a large collection of multimodal datasets provided in the standardized and public MultiBench benchmark (Liang et al., 2021b). This benchmark spans 15 real-world datasets, 10 modalities, 20 prediction tasks, and 6 research areas. Each of these datasets requires a model to learn basic representations of features in each modality and aggregate complementary information across multiple modalities to make a prediction.

**Affective computing** involves understanding our natural display of multimodal signals spanning language (spoken words), visual (facial expressions, gestures), and acoustic (prosody, speech tone) in order to predict human affective states (emotions, sentiment, and personalities) (Picard, 2000). We test on 2 datasets involving fusing *language*, *video*, and *audio* time-series data to predict sentiment and emotions (MOSEI (Zadeh et al., 2018)) as well as humor (UR-FUNNY (Hasan et al., 2019)).

**Healthcare:** Medical decision-making often involves integrating multiple sensory readings from instruments such as lab tests, imaging reports, and patient-doctor conversations (Amisha et al., 2019). We experiment with the large-scale MIMIC dataset (Johnson et al., 2016) which records ICU patient data including *time-series* data measured every hour and other demographic variables in the form of *tabular numerical* data. These are used to predict the disease ICD-9 code and mortality rate.

**Robotics:** Modern robot systems are equipped with multiple sensors in order to capture complementary signals useful for holistic decision-making. We test on the large-scale MuJoCo PUSH (Lee et al., 2020a) and V&T (Vision&Touch) (Lee et al., 2020b) datasets which record the manipulation of simulated and real robotic arms equipped with *visual* (RGB and depth), *force*, and *proprioception* sensors. In MuJoCo Push, the goal is to predict the pose of the object being pushed by the robot end-effector. In Vision & Touch, the goal is to predict action-conditional learning objectives that capture forward dynamics of the different modalities (contact prediction and robot end-effector pose). Both of these datasets include visual, sensor, and contact data collected from real robotics and published at top robotics conferences. Multisensor fusion in robotics is especially important due to the large number of sensors, requirements for robust learning from noisy sensors, and limited data as compared to tasks involving images, text, and audio. From our experiments, our multimodal multitask approach does indeed perform well on these initial robotics tasks, and we do hope to deploy some of these models on real-world robotics as future work.

**Human Computer Interaction (HCI)** studies the design of computer technology and interactive interfaces between humans and computers (Dix et al., 2000). We use the ENRICO dataset (Deka et al., 2017; Leiva et al., 2020) of Android app screens (consisting of an *image* as well as a *set* of apps and their locations) categorized by their design motifs and collected for data-driven design applications such as design search, user interface (UI) layout generation, UI code generation, and user interaction modeling.

**Multimedia:** A significant body of research in multimodal learning has been fueled by the large availability of multimedia data (language, image, video, and audio) on the internet. We experiment on 2 large-scale multimedia datasets with varying sizes and levels of difficulty: (1) AV-MNIST (Vielzeuf et al., 2018) is assembled from *images* of handwritten digits (LeCun et al., 1998) and *audio* samples of spoken digits (Leonard and Doddington, 1993), and (2) CIFAR-ESC (Liang et al., 2021c) is an image-audio retrieval dataset. To construct CIFAR-ESC, we follow Liang et al. (2021c) and combine 100 classes from CIFAR-100 and 10 classes from CIFAR-10 (Krizhevsky et al., 2009) to form 110 image classes, as well as 50 audio classes from ESC-50 (Piczak, 2015). To bridge these two modalities with partially related label spaces, we define 17 shared classes across the 2 datasets for weak concept alignment. These clusters are obtained by mapping similar classes between the datasets using similarities from WordNet (Miller, 1995) and text cooccurrence.

**Multitask setup:** We trained 3 multitask models across combinations of the aforementioned datasets. Each multitask setup is designed to include tasks with different modality inputs and prediction objectives.

1. **Small:** PUSH, V&T: 2 tasks in the same research area (robotics) but with different modality inputs: {image, force, proprioception, control} and {image, force, proprioception, depth} respectively. Furthermore, each robot's sensor readings come from different robot-dependent sensors.

2. **Medium:** ENRICO, PUSH, AV-MNIST across 3 domains (multimedia, HCI, and robotics) with different modalities: {image, set}, {image, force, proprioception, control}, and {image, audio}.

3. **Large:** UR-FUNNY, MOSEI, MIMIC, and AV-MNIST, across 3 domains (affective computing, healthcare, and multimedia), again with different modalities: {text, video, audio} for the first 2 tasks with different format of preprocessed embeddings of video and audio, {time-series, table}, and {image, audio}.

We summarize these experimental settings in Table 1. Overall, the total size of datasets involved in our experiments exceeds 370, 000 and covers diverse modalities such as time-series, various robotics sensors, sets, and tables, as well as multiple research areas and prediction tasks from affective computing, healthcare, multimedia, robotics, and HCI.

## C.2 Hyperparameters, parameter groupings, and training details

We list hyperparameters used throughout our models in Table 8, Table 9, and Table 10 for small, medium, and large multitask settings respectively. Code is also included in the supplementary material for reproducibility.

We train a suite of HIGHMMT models is obtained by tuning $k$, the number of parameter groups (i.e., the number of clusters when clustering heterogeneity matrices). $k$ can be seen as a hyper-parameter depending on the computational budget, with smaller $k$ implying more parameter sharing on lower budgets and vice-versa. We test $k$ in the range $\{2, 4, 6, 7, 9\}$, with $|\mathbb{U}| = 1, |\mathbb{C}| = 1$, $|\mathbb{U}| = 3, |\mathbb{C}| = 1$, $|\mathbb{U}| = 3, |\mathbb{C}| = 3$, $|\mathbb{U}| = 3, |\mathbb{C}| = 4$, and $|\mathbb{U}| = 4, |\mathbb{C}| = 5$ respectively where $|\mathbb{U}|, |\mathbb{C}|$ denote the number of unimodal and crossmodal parameter groups.

More specifically, we choose which modalities to share parameters by selecting the high value entries in the transfer matrix. When $|\mathbb{U}| = 3$, the first set of unimodal parameters are shared betweeen AV-MNIST image and AV-MNIST audio; the second set of parameter is shared between MIMIC table, MOSEI-video and MOSEI-audio, the third set of parameters are shared across the rest of the modalities. When $|\mathbb{U}| = 4$, the first set of unimodal parameters are shared betweeen AV-MNIST image and AV-MNIST audio; the second set of parameter is shared between MIMIC table and MOSEI-video; the third set of parameters are shared between MIMIC time series and UR-FUNNY video, while the last set of parameters are shared between UR-FUNNY text and MOSEI-text.

When $|\mathbb{C}| = 3$, we use two distinct sets of crossmodal parameters for AV-MNIST (image/audio) and MOSEI (video/audio) respectively, while the same crossmodal parameters are shared among the rest of the tasks. When $|\mathbb{C}| = 4$, we share one set between MOSEI (video/audio) and UR-FUNNY (audio/text); one set between UR-FUNNY (video/text) and UR-FUNNY (video/audio); another distinct set for AV-MNIST (image/audio); and finally a set of parameters for the rest. Finally, $|\mathbb{C}| = 5$ is similar to $|\mathbb{C}| = 4$ except that we train a different set of crossmodal parameters for MOSEI (audio/text), and another new set of parameters shared between UR-FUNNY (video/text) and UR-FUNNY (audio-text).

Table 8: Table of hyperparameters for multitask prediction on the **small** setting involving PUSH, V&T: 2 tasks in the same research area (robotics) but with different modality inputs: {image, force, proprioception, control} and {image, force, proprioception, depth} respectively, and readings come from different robot-dependent sensors.

| Part of Model | Hyperparameter | Values | |
|---|---|---|---|
| | | PUSH | V&T |
| Unimodal Perceiver Encoder | Depth | 1 | |
| | Num Latents | 20 | |
| | Latent Dim | 64 | |
| | Cross Attention Heads | 1 | |
| | Latent Self-Attention Heads | 8 | |
| | Cross Head Dim | 64 | |
| | Latent Head Dim | 64 | |
| | Num Latent Blocks Per Layer | 1 | |
| Multimodal Cross-Attention Layer | Depth | 1 | |
| | Num Latents | 20 | |
| | Latent Dim | 64 | |
| | Cross Attention Heads | 1 | |
| | Latent Self-Attention Heads | 8 | |
| | Cross Head Dim | 64 | |
| | Latent Head Dim | 64 | |
| | Num Latent Blocks Per Layer | 1 | |
| Classification Heads (BatchNorm+Linear) | Input/output dimensions | 756/32 | 1280/1 |
| Training | Optimizer | Adam | |
| | Learning rate | 0.0005 | |
| | Weight decay | 0.0 | |
| | Training loss weights | 100.0 | 1.0 |
| | Batchsize | 18 | 64 |
| | Evaluation weights | 100.0 | 1.0 |
| | Original MultiBench Input Dimensions | Gripper Pos: 16x3 Gripper Sensors: 16x7 Image: 16x32x32 Control: 16x7 | Image: 128x128x3 Force: 6x32 Proprio: 8 Depth: 128x128 Action: 4 |
| | Perceiver Input Channel Size | Gripper Pos: 3 Gripper Sensors: 7 Image: 1 Control: 7 | Image: 3 Force: 32 Proprio: 8 Depth: 1 Action: 4 |
| | Perceiver Input Extra Axis | Gripper Pos: 1 Gripper Sensors: 1 Image: 3 Control: 1 | Image: 2 Force: 1 Proprio: 1 Depth: 2 Action: 1 |
| | Perceiver Input Num_freq_bands | Gripper Pos: 6 Gripper Sensors: 6 Image: 6 Control: 6 | Image: 6 Force: 6 Proprio: 6 Depth: 6 Action: 6 |
| | Perceiver Input Max_freq | Gripper Pos: 1 Gripper Sensors: 1 Image: 1 Control: 1 | Image: 1 Force: 1 Proprio: 1 Depth: 1 Action: 1 |
| | Shared Modality Encoding | N/A | |

Table 9: Table of hyperparameters for multitask prediction on the **medium** setting involving AV-MNIST, ENRICO and PUSH: 3 tasks across 3 domains (multimedia, HCI, and affective computing), again with vastly different modality sets: $\{image, audio\}$, $\{image, set\}$, and $\{image, force, proprioception, control\}$ for each task.

| Part of Model | Hyperparameter | Values | | |
|---|---|---|---|---|
| | | AV-MNIST | ENRICO | PUSH |
| Unimodal Perceiver Encoder | Depth | 1 | | |
| | Num Latents | 12 | | |
| | Latent Dim | 64 | | |
| | Cross Attention Heads | 1 | | |
| | Latent Self-Attention Heads | 8 | | |
| | Cross Head Dim | 64 | | |
| | Latent Head Dim | 64 | | |
| | Num Latent Blocks Per Layer | 1 | | |
| Multimodal Cross-Attention Layer | Depth | 1 | | |
| | Num Latents | 12 | | |
| | Latent Dim | 64 | | |
| | Cross Attention Heads | 1 | | |
| | Latent Self-Attention Heads | 8 | | |
| | Cross Head Dim | 64 | | |
| | Latent Head Dim | 64 | | |
| | Num Latent Blocks Per Layer | 1 | | |
| Classification Heads (BatchNorm+Linear) | Input/output dimensions | 128/10 | 128/20 | 768/2 |
| Training | Optimizer | Adam | | |
| | Learning rate | 0.001 | | |
| | Weight decay | 0.0 | | |
| | Training loss weights | 0.8 | 1.0 | 1.1 |
| | Batchsize | 32 | 32 | 32 |
| | Evaluation weights | 1 | 1 | 1 |
| | Original MultiBench Input Dimensions | Colorless Image: 28x28 Audio Spectogram: 112x112 | Image: 256x128x3 Set: 256x128x3 | Gripper Pos: 16x3 Gripper Sensors: 16x7 Image: 16x32x32 Control: 16x7 |
| | Perceiver Input Channel Size | Colorless Image: 16 (cut into 4x4 squares) Audio Spectogram: 256 (cut into 16x16 squares) | Image: 3 Set: 3 | Gripper Pos: 3 Gripper Sensors: 7 Image: 1 Control: 7 |
| | Perceiver Input Extra Axis | Colorless Image: 2 Audio Spectogram: 2 | Image: 2 Set: 2 | Gripper Pos: 1 Gripper Sensors: 1 Image: 3 Control: 1 |
| | Perceiver Input Num_freq_bands | Colorless Image: 6 Audio Spectogram: 6 | Image: 6 Set: 6 | Gripper Pos: 6 Gripper Sensors: 6 Image: 6 Control: 6 |
| | Perceiver Input Max_freq | Colorless Image: 1 Audio Spectogram: 1 | Image: 1 Set: 1 | Gripper Pos: 1 Gripper Sensors: 1 Image: 1 Control: 1 |
| | Shared Modality Encoding | N/A | | |

Table 10: Table of hyperparameters for multitask prediction on the **large** setting involving MIMIC, AV-MNIST, MOSEI and UR-FUNNY: 4 tasks across 3 domains (healthcare, multimedia, and affective computing), again with vastly different modality sets: $\{\text{time-series}, \text{table}\}$, $\{\text{image}, \text{audio}\}$, and $\{\text{text}, \text{video}, \text{audio}\}$ for the final 2 tasks with different format of preprocessed embeddings of video and audio.

| Part of Model | Hyperparameter | Values | | | |
|---|---|---|---|---|---|
| | | MIMIC | AV-MNIST | MOSEI | UR-FUNNY |
| Unimodal Perceiver Encoder | Depth | 1 | | | |
| | Num Latents | 20 | | | |
| | Latent Dim | 64 | | | |
| | Cross Attention Heads | 1 | | | |
| | Latent Self-Attention Heads | 6 | | | |
| | Cross Head Dim | 64 | | | |
| | Latent Head Dim | 64 | | | |
| | Num Latent Blocks Per Layer | 1 | | | |
| Multimodal Cross-Attention Layer | Depth | 1 | | | |
| | Num Latents | 20 | | | |
| | Latent Dim | 64 | | | |
| | Cross Attention Heads | 4 | | | |
| | Latent Self-Attention Heads | 6 | | | |
| | Cross Head Dim | 64 | | | |
| | Latent Head Dim | 64 | | | |
| | Num Latent Blocks Per Layer | 1 | | | |
| Classification Heads (BatchNorm+Linear) | Input/output dimensions | 128/2 | 128/10 | 384/2 | 384/2 |
| Training | Optimizer | Adam | | | |
| | Learning rate | 0.0008 | | | |
| | Weight decay | 0.001 | | | |
| | Training loss weights | 1.2 | 0.9 | 1.1 | 1.5 |
| | Batchsize | 20 | 40 | 32 | 32 |
| | Evaluation weights | 1 | 1 | 1 | 1 |
| | Original MultiBench Input Dimensions | Static: 5 Timeseries: 24x12 | Colorless Image: 28x28 Audio Spectogram: 112x112 | Image: 50x35 Audio: 50x74 Text: 50x300 | Image: 20x371 Audio: 20x81 Text: 50x300 |
| | Perceiver Input Channel Size | Static: 1 Timeseries: 1 | Colorless Image: 16 (cut into 4x4 squares) Audio Spectogram: 256 (cut into 16x16 squares) | Image: 35 Audio: 74 Text: 300 | Image: 371 Audio: 81 Text: 300 |
| | Perceiver Input Extra Axis | Static: 1 Timeseries: 2 | Colorless Image: 2 Audio Spectogram: 2 | Image: 1 Audio: 1 Text: 1 | Image: 1 Audio: 1 Text: 1 |
| | Perceiver Input Num_freq_bands | Static: 6 Timeseries: 6 | Colorless Image: 6 Audio Spectogram: 6 | Image: 3 Audio: 3 Text: 3 | Image: 3 Audio: 3 Text: 3 |
| | Perceiver Input Max_freq | Static: 1 Timeseries: 1 | Colorless Image: 1 Audio Spectogram: 1 | Image: 1 Audio: 1 Text: 1 | Image: 1 Audio: 1 Text: 1 |
| | Shared Modality Encoding | The text modality from MOSEI and UR-FUNNY are shared. | | | |

Table 11: We train multitask HIGHMMT on 1/2/3 datasets and find that it generalizes to new modalities and tasks on the 4th dataset, with improved performance over single-task training on the 4th dataset. 0 source tasks implies transferring randomly initialized parameters, which is equivalent to single-task training on the target task. Cross-modal transfer improves with the number of pretraining tasks and works best on the smallest target tasks (UR-FUNNY).

| Source tasks | Target task UR-FUNNY |
| --- | --- |
| 0 (no transfer) | 63.1 ± 0.5 |
| MOSEI | 63.5 ± 0.5 |
| MOSEI + AV-MNIST | 64.0 ± 0.7 |
| MOSEI + AV-MNIST + MIMIC | **64.7 ± 0.4** |

| Source tasks | Target task MOSEI |
| --- | --- |
| 0 (no transfer) | 79.0 ± 0.5 |
| AV-MNIST | 79.2 ± 0.3 |
| AV-MNIST + MIMIC | 79.3 ± 0.5 |
| AV-MNIST + MIMIC + UR-FUNNY | **79.6 ± 0.6** |

| Source tasks | Target task MIMIC |
| --- | --- |
| 0 (no transfer) | 67.7 ± 0.6 |
| MOSEI | 67.9 ± 0.5 |
| MOSEI + AV-MNIST | 68.0 ± 0.8 |
| MOSEI + AV-MNIST + UR-FUNNY | **68.4 ± 0.6** |

| Source tasks | Target task AV-MNIST |
| --- | --- |
| 0 (no transfer) | 70.3 ± 0.4 |
| MOSEI | 70.5 ± 0.4 |
| MOSEI + MIMIC | 70.5 ± 0.4 |
| MOSEI + MIMIC + UR-FUNNY | 70.6 ± 0.4 |

# D  Additional Results

In this section, we detail additional experimental results that support the multitask, transfer, and generalization capabilities of HIGHMMT.

## D.1  Generalization to new modalities and tasks

HIGHMMT also offers opportunities to study whether we can *transfer* knowledge between completely different tasks and modalities. On the large setting, we first pretrained a model on 0/1/2/3 of the four tasks before fine-tuning on the fourth task only. We show these full results in Table 11. On all four target tasks, our proposed multitask pretraining and fine-tuning paradigm improves performance over single target-task training. Therefore, weights learned from other multimodal tasks indeed generalize well to new modalities and tasks. We further analyze this transfer learning phenomenon by studying the following research questions:

**Effect of pretraining datasets.** When we vary the number of pretraining datasets, we observe a consistent improvement on fine-tuned target task performance across all datasets. This effect is particularly pronounced on the UR-FUNNY target task, which shows the biggest improvement using pretrained parameters from 0 to 3 multitask datasets. This implies that HIGHMMT learns more generalizable multimodal features as more tasks are involved in multitask training.

**Effect of target dataset size.** We observed an inverse correlation between target task size and performance improvement: the smallest dataset, UR-FUNNY, benefited the most (+2.4%) from transfer learning. This implies that this multimodal pretraining-fine-tuning paradigm is useful for improving performance for low-resource target modalities and tasks.

**Effect of transfer modalities.** We compare transfer learning performance across different levels of partial observability. While one would expect transfer to the MIMIC dataset to be the hardest due to its modality

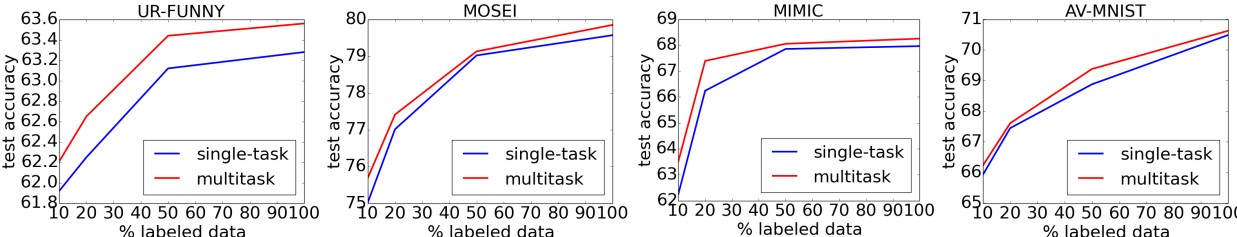

Figure 9: **Few-shot results** on new modalities and tasks. Multimodal multitask training using HIGHMMT learns more generalizable representations which improves performance across all ranges of data. The $x$-axis shows the percentage of labeled data used during training.

Table 12: Multitask HIGHMMT also enables training a single model for both multimodal fusion and retrieval tasks.

| Model | AV-MNIST ↑ | CIFAR-ESC ↑ | Params (M) ↓ |
|---|---|---|---|
| HIGHMMT | 70.4 | 58.8 | 1.04 |
| HIGHMMT multitask | 70.4 | **60.5** | **0.52** |

set {time-series, table} being completely disjoint from the remaining 3 datasets, we still observe a +0.8% gain as compared to single-task training. Therefore, HIGHMMT can generalize to new modalities and tasks. Unsurprisingly, for datasets with more overlap in modality sets (e.g., UR-FUNNY with complete overlap in {text, video, audio} as compared to the other 3 datasets used for pretraining, we find larger improvements using transfer learning over single-task models (+2.4%).

**Comparisons to unimodal transfer.** Recent work has explored the possibility of transferring Transformer representations trained in one modality to another Lu et al. (2021); Kiela et al. (2019). Our transfer experiments also corroborate these findings in a multimodal setting with promising results on new modalities and tasks, especially involving real-world, smaller, and noisier datasets such as those involving human videos (MOSEI and UR-FUNNY), medical data (MIMIC), or real and simulated robots (PUSH and V&T).

## D.2 Few-shot learning

HIGHMMT offers opportunities for statistical strength sharing across tasks. We test this hypothesis in the few-shot learning scenario, by evaluating whether multitask information sharing can improve performance on low-resource target tasks. We compare a single-task HIGHMMT trained only on a percentage $p$ of labeled training data in the target task with multitask HIGHMMT trained on the same percentage $p$ (during multitask training we prioritize performance of the target task over others). By varying $p \in [0.1, 1.0]$, we plot the performance under few-shot settings in Figure 9. We find that multitask training is consistently better across all ranges of data, which supports the fact that more generalizable representations across modalities and tasks are learned in HIGHMMT. The main takeaway is that if it is too difficult to collect data in a target domain, collecting data from a different domain and using a shared multimodal model is an alternative approach for improving performance.

## D.3 Multitask fusion and retrieval

To assess task generalization, we train multitask models over fusion in AV-MNIST and retrieval in CIFAR-ESC. While fusion emphasizes information integration from complementary data sources, retrieval focuses on aligning corresponding elements expressed through different views of the data (Baltrušaitis et al., 2018). Table 12 shows the full results of this experiment: even across vastly different multimodal prediction tasks, we find that multitask training (60.5% retrieval accuracy) improves upon single-task training (58.8% accuracy), while performance on the AV-MNIST fusion tasks is similar for both single-task and multitask learning. Not only have the unimodal encoders simultaneously processed different modalities, the multimodal attention layer has also learned to capture correspondences useful for both fusion and retrieval, while halving the total number of parameters required as compared to task-specific modeling.

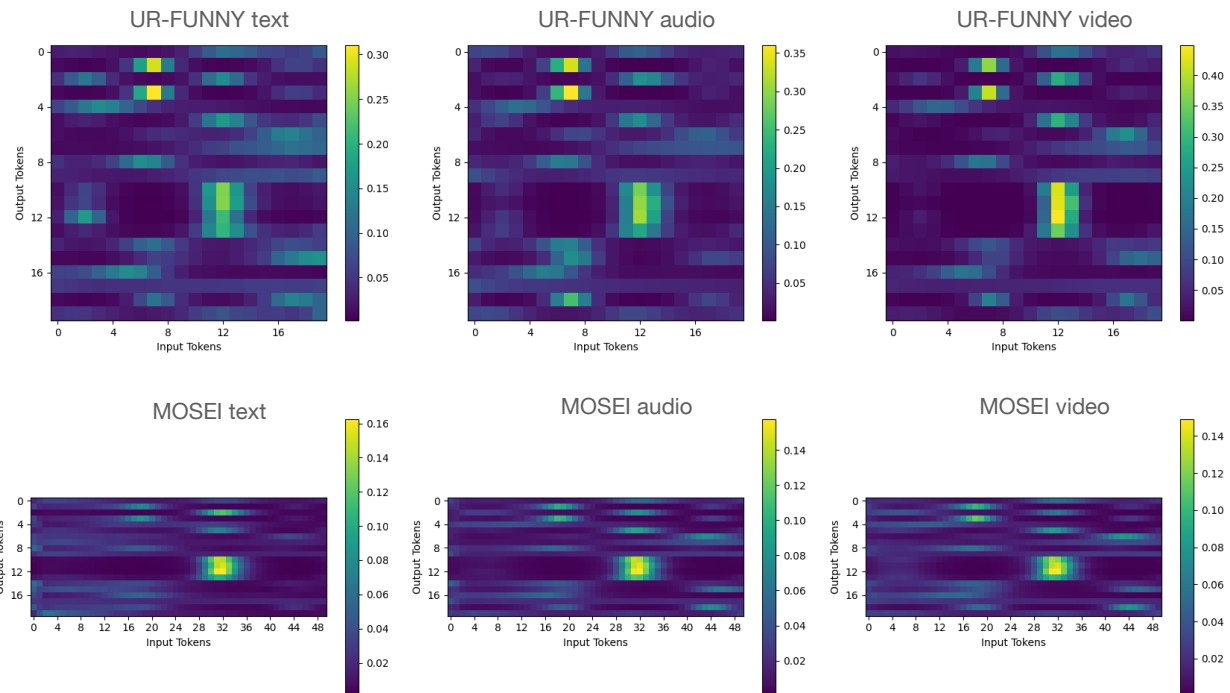

Figure 10: Visualizations of attention patterns learned by the unimodal encoders across different modalities in the UR-FUNNY and MOSEI datasets, which consist of text, audio, and video modalities aligned in time. We find that there are some common attention patterns across tasks, which implies that the shared unimodal encoders have learned common information across tasks.

## D.4 Understanding HighMMT

In this subsection, we analyze why this general model achieves strong results in multitask, transfer, and few-shot settings. Based on prior work in multitask learning (Caruana, 1997; Ruder, 2017; Zhang and Yang, 2021), we set up two possible hypotheses: (1) improved generalization and (2) improved regularization. We supplement the results in the main paper with additional visualizations and comparisons in this subsection.

### D.4.1 Hypothesis 1: Improved generalization

**Investigating parameter sharing.** In which components of the HighMMT model is parameter sharing important?

We further study the importance of parameter sharing in HighMMT. From the ablation studies in Table 5, using separate parameters for either unimodal or multimodal layers results in worse performance. The full model with completely separate unimodal and multimodal layers is reminiscent of typical single-task multimodal transformers (Tsai et al., 2019; Lu et al., 2019; Hendricks et al., 2021) trained separately for each task. We show that HighMMT maintains competitive performance (with slightly better performance on several datasets) due to statistical strength sharing, while also reducing parameters by 6× due to sharing of unimodal encoders and multimodal layers across tasks.

Furthermore, we surprisingly find that removing the modality-specific embedding layer results in only slightly worse performance (70.3 to 69.6 average score). This implies that the shared unimodal encoder has learned generalizable feature extractors that can encode heterogeneous modalities even without a modality identifier.

**Visualization of attention patterns.** How do the shared unimodal encoders attend to modality-specific tokens?

Given that parameter sharing seems to be useful for performance and efficiency, we aim to better visualize the nature of information sharing in the attention layers of unimodal encoders. We perform inference on a

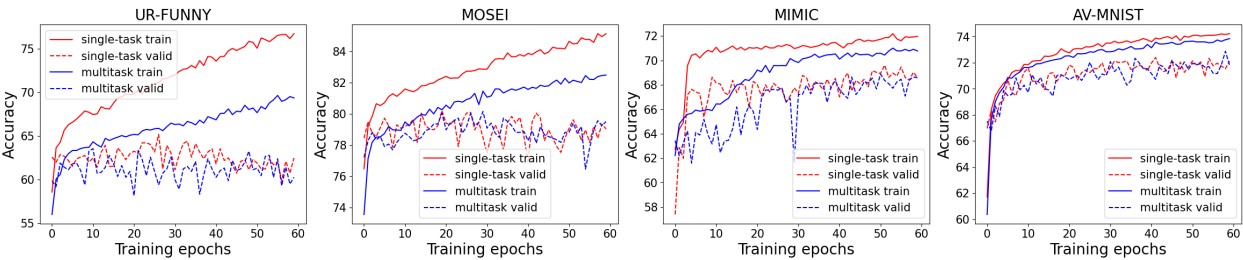

Figure 11: Multitask models converge as fast but overfit less (a smaller gap between train and valid accuracies) vs single-task models, which implies that multitask training helps to regularize the joint parameters and reduces overfitting on the target task.

trained multitask HIGHMMT model on the test data of the large multitask setting, and average the attention patterns across test datapoints for each dataset. Following Lu et al. (2021), the average attention pattern provides information on general inductive biases captured by the unimodal encoders and enables us to make holistic conclusions rather than comparing attention maps on individual datapoints.

From Figure 10, we actually find that a common attention pattern emerges across modalities and tasks. First looking across modalities in the same dataset, we find that the model captures common temporal patterns at the same time steps, which makes sense since the 3 modalities are time-aligned in a video. The attention patterns are quite similar which implies that the same attention strategy can often work well across different modalities and tasks. This could be an explanation of why our model is able to perform multiple tasks simultaneously using shared parameters in attention layers.

It is also interesting to see how the model automatically learns to "divide up work" amongst its 20 latent tokens (numbered 0-19): the latent tokens $10 - 13$ typically all focus on the region about two-thirds after the start of the input sequence, while latent tokens 1 and 3 always focuses on the region about one-third from the start. Certain tokens (3 and 18) seem to learn oscillating attention patterns, and certain pairs of tokens learn complementary attention patterns (e.g., 4, 5, and 6 attend one after the other). There are also some latent tokens that more evenly attend to the whole input sequence, such as latent tokens 9 and 17, which can be seen as "summary" tokens. This shows that the perceiver-based encoder is able to divide up its limited latent space well to capture important information both in specific time-steps and contextual information across all the time-steps, thus creating a holistic representation of the input using a much smaller set of latent variables.

### D.4.2 Hypothesis 2: Improved regularization

In parallel to improved generalization, another line of research has focused on the regularization effects of multitask learning. Baxter (1997) showed that multitask parameter sharing reduces the risk of overfitting on the original task by forcing the model to learn across multiple tasks. We study the following regularization effects:

**Training dynamics.** In Figure 11, we traced the train and valid accuracies across 60 training epochs (with multitask training in the large setting). The training process of HIGHMMT converges at about the same rate between single-task and multitask learning, but the multitask model overfits less (a smaller gap between training and valid accuracies). This implies that multitask training helps to regularize the joint parameters and alleviates their overfitting on the target task.

**Task weights.** We found that optimizing a simple weighted sum of loss functions over all tasks was sufficient to obtain strong multitask performance. Instead of assigning uniform weights to each task, sometimes we found it helpful to set the weight higher for more challenging datasets during HIGHMMT multitask training. We show some examples of this phenomenon in Figure 12, where multitask performance can sometimes be sensitive to weights especially when prediction objectives are of different scales (i.e., MSE vs accuracy). This supports the regularization argument where carefully tuned weighted auxiliary objectives encouraging the model to also fit other auxiliary tasks can help improve performance on a target task. However, doing so would not achieve the best performance on auxiliary tasks.

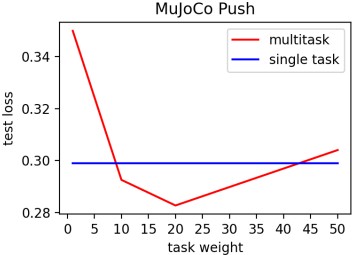 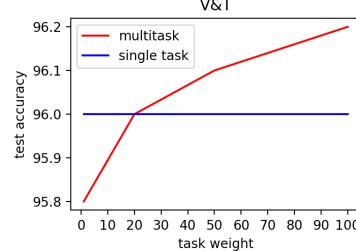

Figure 12: Multitask performance can sometimes be sensitive to task weights especially when prediction objectives are of different scales (i.e., MSE for PUSH vs accuracy for V&T), in a manner similar to how carefully-tuned regularization terms help in training models.

### D.5   Summary of main take-away messages

In conclusion, we designed a general multimodal multitask model for high-modality (a large set of diverse modalities) and partially-observable (each task only defined on a small subset of modalities) scenarios. Our approach relies on training for *multitask* and *transfer* learning: multitask learning with shared unimodal and multimodal layers enables stable parameter counts (addressing scalability) and cross-modal transfer learning enables information sharing across modalities and tasks (addressing partial observability). Through an extensive set of experiments and analysis, we summarize our main take-away messages as follows:

1. **Standardized multitask modeling.** We train a single multitask HIGHMMT model for numerous high-modality and partially-observable multimodal tasks (across 10 modalities, 15 prediction tasks, and 5 research areas), achieving strong performance while reducing total parameter counts. We believe that standardized modeling leads to a smaller set of architectural decisions, enables transfer to understudied modalities and tasks, and present a unified platform for subsequent theoretical and empirical analysis.

2. **Cross-modal transfer to new modalities and tasks.** Multitask HIGHMMT enables cross-modal information transfer by pretraining on source multimodal tasks before transferring to completely new target modalities and tasks. Involving more tasks during pretraining improves performance, and gains are more apparent when fine-tuning on low-resource target tasks. This finding can supplement current pretrain-finetune paradigms typically performed on the same modality (e.g., text-only or image-only), and encourage research in more general multimodal pretraining over high-modality settings before fine-tuning on only a partial subset of all observed modalities.

3. **Tradeoff between performance and efficiency.** Multitask HIGHMMT improves the tradeoff between performance and efficiency over task-specific state-of-the-art models especially in low-resource scenarios (less training data and partially-observable modalities). Coupled with the relatively fewer architectural decisions and generalization to understudied modalities and tasks, we believe that multitask HIGHMMT and similar architectures should be a starting point for future research.

4. **Few-shot multitask learning.** Multitask information sharing can improve performance on low-resource target tasks with limited labeled training data. Therefore, if it is too difficult to collect data in a target domain, collecting data from a different domain and using a shared multimodal model is an alternative approach for improving performance.

5. **Information sharing.** Finally, our analysis reveals surprising insights regarding the nature of information sharing in multimodal and multitask models, which may be of independent interest. Specifically, there are both generalization and regularization effects at play:

   - On the generalization side, information sharing is present across modalities and tasks, but at different levels across shared unimodal and multimodal layers. Information sharing enables strong multitask performance even under partial-observability and generalization to new modalities and tasks.

   - While using modality-specific embeddings achieves the best performance, there is only a minor drop when removing them, which implies that shared unimodal encoders can learn generalizable feature extractors even without a modality identifier.

   - On the regularization side, well-tuned regularization weights yield training dynamics that display less overfitting on target tasks as compared to single-task learning.

