# OpenReview forum: "High-Modality Multimodal Transformer: Quantifying Modality & Interaction Heterogeneity for High-Modality Representation Learning"
_TMLR — Accepted by TMLR_

### Review · Reviewer_tic5 · 2023-01-27

**Summary Of Contributions:**

This paper addresses the multimodal learning topic by the standpoint of knowledge transfer, i.e., trying to investigate how several combinations of (even very different) data modalities and tasks behave so to improve generalization and transfer capacity to new diverse modality/task. This work specifically considers under-investigated modalities (such as, e.g., tabular data) and proposes to measure the heterogeneity between the modalities, so to assess the similarity of the encoded information as well as of the interaction, in order to allow parameter sharing. Hence, two information theoretic metrics to measure modality and interaction heterogeneity are proposed, in order to quantify, let's say, the "affinity" (in terms of information transfer) between two modalities and pairs of modalities (for interaction).
Subsequently, a new model, named HighMMT, is presented, able to deal with diverse combination of modalities and tasks, regardless the type of data, cardinality, and other peculiar characteristics of the data and of the tasks. A large experimental analysis is reported considering the MultiBench benchmark, which considers text, images, audio, video, non-visual sensors, proprioceptive data, speech, time series, sets and tables, showing promising results in terms of performance (the model provides higher accuracies as long as new modalities are added, and transfer well to new modalities and tasks during fine tuning) and efficiency (scales well due to the parameter sharing).


**Audience:**

Yes

**Broader Impact Concerns:**

A section about  Broader Impact Statement is not present, but I don't see ethical concerns for this work, at least no more than other machine/deep learning work of the same flavor.

**Claims And Evidence:**

No

**Requested Changes:**

The main issues regard clarity and organization of the content of the work. Meaningful information is in the appendix and should be included in the main paper, but especially, practically all the paper should be rewritten in order to clarify and justify the proposed method and the several stages.
Many of the specific issues have been reported above, and should be referred for the paper revision.

AFTER REBUTTAL

I believe that the paper, even after authors' revision, is still affected by notable problems to be solved, first of all regarding the organization and clarity of the work, which cannot only be fixed by adding paragraphs here and there along the manuscript.
For example, the modality and interaction heterogeneity measures seen under the perspective of information transfer are not enough convincing: they both look very similar considering the difference of the losses between standard training with random init weights and training using init weights taken from former pre-training with the other modality, and this is valid for both single modalities and pair of modalities. I don't see here a convincing justification of such an approach and, moreover, the method is combinatorial with the number of modalities, so poorly scalable.

Assumption of distances as metrics only valid for some percentage of the time remains fuzzy.

Experimental results suffer of the same problem: despite the explanation is a bit better, I still deem that given the complexity of the method is difficult to understand which its actual contributions, and results do not seem significantly better in a statistical sense.

Overall, I acknowledge that authors made a lot of work in proposing this methodology, but the way it is presented should be much improved to be made understandable, convincing and acceptable for publication.


**Strengths And Weaknesses:**

This work addresses interesting topics, not so well explored in the multimodal learning literature, as, for instance, the "similarity" between different modalities and the contribution of diverse multimodal data/tasks in the performance of a very different new task, but also there are several shortcomings.
In general, the paper is far from being clear, it is difficult to catch the technical aspects of the work, just for the way it is presented. The wording is in some cases hard to understand and hard to figure out the message authors want to convey. Moreover, some of these contributions are not properly justified, and it is difficult to find an accurate, convincing and clear evidence of the stated claims.

First of all, as per the proposal of the information theoretic metrics for heterogeneity measurements of the modalities, I haven't understood clearly the meaning of the interaction heterogeneity, that is, the transfer capability of a pair of modalities to another pair. While it is natural to assess the heterogeneity between 2 modalities, this concept of "interaction" is not explained, and indeed I found difficult to understand the utility. One could also ask why modality pairs and also triple or more modalities.
The second contribution relates the proposal of a single model able to manage the large amount of multimodal data and tasks, namely, 10 modalities, 15 tasks in different research areas. While the use of a "single" model is surely an interesting aspect, the main paper develops without knowing the type of model proposed, other than it is likely a transformer-like system; the actual architecture is only presented in the Appendix, so one attempts to understand everything without knowing how this model manages different data (size, cardinality) and tasks, and their combination, how information flows, etc.. Moreover, apart some contributions (not original) about the information theoretical metrics, it is hard to find any theoretical ground justifying the approach: this is not necessarily a big issue, but accompanied with the pure procedural approach, made the proposed methodology a bit weaker.

More specifically, the contribution about the metrics is firstly reported, it derives from a variation of mutual information ($\mathcal{V}$-usable information) presented in former works, exploited here to assess the similarity/difference (called heterogeneity) between 2 modalities (and pairs of modalities). The derived theory is presented cast to the problem at hand, but the conclusion derives from an empirical observation, not corroborated by any other support, but just saying that experiments evidence that $\mathcal{V}(X_1)$ and $\mathcal{V}$ are both expressive enough to allow a simplification in formula (3). Moreover, the introduction of the distance $d$ is claimed to respect the metric rules (non-negativity, symmetry, and triangle inequality), but this is also not demonstrated.
The same reasoning is applied to interaction heterogeneity measurements, but still issues of clarity are present.

This analysis is then used to define heterogeneity among the modalities and determine parameters' grouping in the subsequent model bia hierarchical clustering, which is not well addressed too, just reporting that the number of clusters $k$ is a hyper-parameter depending of the computational budget. hence, the proposed model presented in the Appendix is subject to homogeneous pre-training across all modalities followed by an heterogeneity-aware fine tuning.
The explanation for the pre-training stage goes through a set of steps addressing the standardization of the input sequences (across modalities) and further data management and training of unimodal and cross-modal networks up to final classification stage, which can be task-specific or multitask. The heterogeneity-aware fine tuning looks for the similarity of the modalities and pairs of modalities and groups accordingly the networks' parameters, and, starting from the homogeneous pre-trained models, separately fine tune the model.
Not so clear explanations and justifications are given for all these steps as it is likely evident from my description above. The procedural stages are not better described, which results difficult from a reader to fully catch why this procedure is followed. This is probably true for readers not particularly skilled in Transformers, but I think that also Transformers experts can find problems to fully understand the methodology adopted as reported in the paper.

As per the experimental session, I found similar issues. For instance, datasets are quoted without explaining what they are, leaving such description still in the Appendix. Some comments are here reported but it resulted difficult to figure out well since the datasets and tasks were not explained in advance. In short, the overall section suffers from the same problems of clarity present also in the previous sections, as well as in claims not clearly supported by the results.
For instance, Table 1 shows some quantitative performance on the transfer to new tasks, but the improvement is in some cases not so evident, given all the expectations of the method (it is claimed about "strong" evidence, but it is indeed not so strong).
Table 2 presents other results wrt the sate of the art, without actually mentioning the works they are comparing to.
Also the ablation studies are not so clearly explained and figures are sometimes contradictory (see Table 3, e.g., direct training re HighMMT results). Finally, all results are given as single performance, but it'd safer to validate the model with more runs per single experiment and report results with average and standard deviation (variance) figures.

State of the art is quoted at the end of the paper, but it results difficult to cast the proposed work in relation to it, and, as mentioned above, comparisons in terms of performance is not well addressed.

The Appendix improves a bit the situation but still issues about clarity remain. For example, Fig. 7  shows the HighMMT architecture but it's unknown where the latent vector in input comes from. Other claims are reported in other sections in the appendix but they are not well corroborated by evidence or justified properly, e.g., the low-rank nature of the modality and interaction heterogeneity matrices, the use of Fourier transformation for positional encoding.
Moreover, going into more details regarding the procedure, specifically the data standardization step, since all data should be normalized across modalities taking the max dimensions, it results that AV-MNIST data are very low dimensional and require zero-padding of 336 out of 387 elements: this sounds a bit odd to me, and would require better investigation. Other specific steps of the training procedure would require more explanation and, likely, further ablation analyses.
Finally, Fig. 10 reports the visualization of attention patterns learned by the unimodal encoders across different modalities and data, and it's claimed that such visualizations evidence common (same position) patterns across tasks, but this finding is not so straightforward to me: given the variety of the data modalities and tasks, I'd expect that each encoder catches an attention pattern for each modality which is not necessarily in the same position of other data modalities, rather, I'd expect different patterns in dependence of the specific modality and task.

In the end, poor paper organization (including both technical stuff and experiments), clarity of the exposition, claims not always corroborated by evidence, and sometimes too strong, poor theoretical ground, and also specific technical issues affect the quality of this work.

---

> ### Author Response · Authors · 2023-03-05
> **Thank you for your feedback, we have revised our paper accordingly - part 1**
>
> Thank you for your valuable feedback and insightful comments! We respond to some concerns below:
>
> > Meaning of interaction heterogeneity
>
> **We have added these necessary details in Section 2.1.** Modality heterogeneity occurs because the information present in different modalities often shows diverse qualities, structures, and representations. Interaction heterogeneity occurs because different modalities interact differently to give rise to new information when used for task inference. In a setting with many modalities, there are many choices on how to encode each modality and the fusion between modalities. Without properly understanding the differences in modalities and their interactions, we will either not be able to achieve the best performance (i.e., overly general models), or will end up using many additional parameters as the number of modalities and tasks increases (i.e., overly specific models). Measuring modality heterogeneity enables us to answer: what is the right unimodal encoder for $X_1$ and $X_2$? Should it be the same, or two different ones? Measuring interaction heterogeneity enables us to answer: what is the right fusion model to fuse $\{X_1,X_2\}$ and $\{X_3,X_4\}$? Should it be the same, or two different ones? Together, measuring heterogeneity gives a principled approach to design models that prioritize the fusion of modalities that contain unique information or interactions using separate parameters.
>
> > One could also ask why modality pairs and also triple or more modalities.
>
> It is definitely useful to explore interaction heterogeneity over 3 or more modalities, but the computational complexity grows very quickly. Our work takes an initial step to demonstrate the usefulness of heterogeneity-aware general models, and the exploration of higher-order heterogeneity is ripe for future work. There are also architectural questions since current multimodal transformers are designed to only capture cross-modal attention between 2 modalities at a time, so new modeling approaches are future directions as well.
>
> > The main paper develops without knowing the type of model proposed, other than it is likely a transformer-like system; the actual architecture is only presented in the Appendix, and the procedural stages are not better described.
>
> We apologize for omitting the model architecture details from the main paper. **We have added a substantial discussion of the underlying multimodal model in Section 2.2 using mathematical notation and relevant figures**. You are right that it is a multimodal transformer model which builds on top of the Perceiver architecture, so the same latent dimension is used for all input modalities for efficiency. We also extend it into a cross-modal Perceiver using cross-attention mechanisms on top of unimodal representations to learning multimodal features (see equation 11). We found that the crossmodal extension was necessary from our ablation studies (Table 5). Thereafter, with this architecture fixed, we explore parameter sharing and splitting across the unimodal and crossmodal parameters using heterogeneity quantification.
>
> > V(X1) and are both expressive enough to allow a simplification in formula (3).
>
> This is a general property of function approximators like neural networks, which Xu et al., (2019) showed empirically. Both $H_{\mathcal{V}(X_1)} (Y)$ and $H_{\mathcal{V}} (Y)$ reduce to the label entropy since the entropy over $Y$ is computed without any input to the model, so it does not matter what the model parameters are initialized with (random or transferred from training on $X_1$): both models just fit the label distribution immediately and reduce to $H(Y)$. Anyways, following the suggestion of reviewer Yccr, we changed the motivation from V-information to a much simpler one, through ‘how much pretraining on one modality helps transfer to another’, which alleviates some of these issues. **We have updated section 2.1 substantially to make the description of heterogeneity via pretrain-transfer much clearer.**
>
> > Distance d metric rules
>
> **Following our new descriptions, we have also clarified the scenarios when such a distance equation satisfies metric properties and provided proofs in Appendix A.1**. We do need to make some assumptions to ensure that the distance is exactly a metric, and these assumptions may not hold in practice since the exact properties of the pretrain-finetune paradigm remains a big open question in modern ML. We have included detailed discussions of this in Appendix A.1 and showed that these properties (e.g., triangle inequality) hold >90% of the time in practice, which enable low-rank approximations of the heterogeneity matrices for computational efficiency. Our method also works well in practice enabling positive transfer by pretraining on different modalities and tasks for downstream modalities and tasks.

---

> > ### Author Response · Authors · 2023-03-05
> > **Thank you for your feedback, we have revised our paper accordingly - part 2**
> >
> > > Parameter grouping via clustering
> >
> > The number of clusters k is a hyper-parameter depending on the computational budget. We have added **Table 2 to show a comparison** between the number of clusters chosen, the number of parameters the resulting model is, and the average performance. More clusters lead to more separate parameters which can boost performance but at the expense of greater complexity.
> >
> > > Datasets are quoted without explaining what they are, leaving such description still in the Appendix.
> >
> > **We have now added a table summarizing the datasets we use and their input modalities, datapoints, and prediction tasks in Table 1**. Due to lack of space the remaining details are in the Appendix. We also plan to release all code and datasets for this paper (and it is already attached on Openreview as supplementary material), so readers can fully explore and continue using these datasets.
> >
> > > For instance, Table 1 shows some quantitative performance on the transfer to new tasks, but the improvement is in some cases not so evident
> >
> > We would like to emphasize that Table 3 (Table 1 in the previous version) reports few-shot transfer results: training on a set of modalities and few-shot transferring to a completely new set of modalities, which, to the best of our knowledge, nobody has been able to achieve in multimodal settings. Lu et al. (2021) do it only for unimodal transfer from language models to other modalities. The key here is that pretraining on other modalities demonstrates a critical scaling behavior - target task performance increases as more source modalities and tasks are used during pretraining, which presents a strategy for pretraining even if target modalities and tasks are not abundant. In light of this new experimental setting, its potential impact, and lack of existing results, we believe that these results are strong and present many avenues for future work in scaling them up.
> >
> > > Table 2 presents other results wrt the state of the art, without actually mentioning the works they are comparing to.
> >
> > All Table 4 (previously Table 2) SOTA results are obtained by exhaustively testing a suite of more than 20 task-specific models, and **we have now updated the table to include names and references** to these of top-performing models. All these models are well-established multimodal architectures summarized in Liang et al., (2021b), spanning innovations in fusion architectures (e.g., early, late, attention, multiplicative interactions, tensor gated, kernel, transformer, etc.), optimization objectives (reconstruction, self-supervised learning, contrastive learning, alignment, robust optimization etc.), and training approaches (balancing training). These methods are constantly improving, and we used the latest versions of each method. We believe these are extremely strong baselines spanning both breadth and depth of current task-specific multimodal methods.
> >
> > > Ablation studies are not so clearly explained and figures are sometimes contradictory (see Table 3, e.g., direct training re HighMMT results).
> >
> > We have clarified in a bullet-point list the exact descriptions of each ablation study. We also apologize for the potentially ambiguous wording of direct training - we used direct training to refer to training HighMMT directly according to the learned parameter groups, whereas normal HighMMT training was performed by pre-training all parameters together, before fine-tuning into separate parameter groups. To increase clarify we have changed direct training to w/o homogeneous pre-training, which clarifies the key missing component in this ablation as compared to full HighMMT training. Table 5 (previously Table 3) indeed shows that removing the homogeneous pre-training stage hurts performance since it quickly overfits to smaller datasets.
> >
> > > Validate the model with more runs per single experiment and report results with average and standard deviation.
> >
> > **We have updated all numbers in Tables 2, 3, 4, and 5** which are our core numerical results with mean += standard deviation over 10 runs. Our key multitask, transfer, and ablation results still hold.

---

> > > ### Author Response · Authors · 2023-03-05
> > > **Thank you for your feedback, we have revised our paper accordingly - part 3**
> > >
> > > > State of the art is quoted at the end of the paper, but it results difficult to cast the proposed work in relation to it, and, as mentioned above, comparisons in terms of performance is not well addressed.
> > >
> > > **It is perhaps easiest to see overall performance in Figure 6 and descriptions at the bottom of page 8, where we compare to more than 20 models in each task for a total of >10,000 model combinations total across all 7 datasets**. HighMMT share all (dark green) is the most parameter efficient, results in a model that reaches within 96% of current state-of-the-art performance but using 30× fewer total parameters. Optimizing for performance results in a model as strong as current state-of-the-art models while using 8× fewer total parameters. The suite of sharing heterogeneity-aware HighMMT (in yellow) all push forward the Pareto front, achieving significantly better tradeoffs between performance and efficiency.
> > >
> > > We have also added **additional results and references in Table 5** to indicate which of the ablations recover recently proposed task-general approaches that concatenate all modalities on the sequence dimension (e.g., Gato) or those using shared cross-attention (e.g., FLAVA) to train a single shared model (without heterogeneity-quantification) for image and text tasks.
> > >
> > > > Fig. 7 shows the HighMMT architecture but it's unknown where the latent vector in input comes from.
> > >
> > > The latent vector is randomly initialized and trainable, we have **edited figure 3** (previously in the Appendix, now moved to the main paper) to illustrate this. Each latent vector $Z_i$ can be seen as a representation initialized randomly and accumulating information from each modality $X_i$ through cross-attention from $Z_i$ to $X_i$ and self-attention from $Z_i$ to itself. Summarizing all information into a common latent array $Z_i$ regardless of the input shape results in total runtime only linear with respect to the input size (rather than quadratic in conventional self-attention on the input) which scales to high-modality scenarios.
> > >
> > > > Low-rank nature of the modality and interaction heterogeneity matrices.
> > >
> > > We have added experiments on this, showing the magnitude of eigenvalues after SVD on the matrices in Appendix A.2. Therefore, using a low-rank approximation of r = 3 is sufficient to approximate the entire matrix, so running a random sample of $O(M)$ pairs of heterogeneity values, and imputing the rest of the heterogeneity matrix, is sufficient in practice. **We have updated section 2.1 with a discussion of complexity and section 3.1 with additional details on this approximation.**
> > >
> > > > Fourier transformation for positional encoding
> > >
> > > We directly follow prior work standard in sequence models, Transformers, and Perceiver. These Fourier features directly represent the position structure of the input data (preserving 1D temporal or 2D spatial structure for audio or images.
> > >
> > > > AV-MNIST data are very low dimensional and require zero-padding of 336 out of 387 elements: this sounds a bit odd to me, and would require better investigation.
> > >
> > > We would like to clarify that AV-MNIST is not low-dimensional, its visual modality (which needs 336 dimensions of padding) are 28x28 images, thus they are 784 dimensional inputs. However, directly inserting a 784-dimensional vector as input to Perceiver does not make sense as (1) it will greatly enlarge the maximum channel size and thus require much more padding from all other modalities; and (2) it does not fit common practice for inputting visual data into transformers. Vision transformers usually cut up the 28x28 image into smaller 4x4 square patches and input the patches sequentially to the transformer, resulting in a transformation from a 784-dimensional static vector into a sequence of length 49, each of dimension 4x4=16. Some other modalities (e.g., text) have a feature dimension of 300, so the feature dimension is eventually padded up to 387 (after adding positional encodings and modality embeddings), and sequence length can be variable (ranging from 49 for AV-MNIST images to >100 for text, videos, time-series, and sensors). In practice, we pass the list of padded indices into PyTorch so that the forward pass and gradient backpropagation are only computed on the non-zero padded indices, so padding will not affect final model predictions.
> > >
> > > > Visualization of attention patterns
> > >
> > > For the attention patterns we showed, the 3 images in the top row represent 3 modalities (text, audio, video) aligned in time for a common task, so there is a common time dimension (x-axis) which the model commonly uses to capture modality-shared information. There is also modality-specific information since the attention patterns are different across modalities for the same task: many more patches are activated in UR-FUNNY text but not UR-FUNNY video, and activated in MOSEI video but not MOSEI text. There are also modality-specific embeddings appended to the input which can capture modality-specific information.

---

> ### Author Response · Authors · 2023-03-24
> **Author response to review after rebuttal - part 1**
>
> Dear reviewer, thank you so much for looking through our response and updating your review. We would like to clarify the following points:
>
> > Organization and clarity of the work.
>
> We would like to emphasize that we really really appreciated your feedback regarding clarity of the model architecture and experiments, and we have tried our absolute best to incorporate them in our revisions. We did not simply 'add paragraphs along the manuscript sections', but rather a complete rewrite of Section 2.1 to make the description of heterogeneity via pretrain-transfer much clearer, added discussion of computational complexity, added a completely new discussion of the underlying multimodal model in Section 2.2 using mathematical notation and relevant figures, updated Tables 2, 3, 4, and 5 to clarify the experimental results, and added Appendix 1 and 2 to clarify the properties of our heterogeneity estimators. Finally, we took a pass to ensure the flow of the entire paper and made more edits. If there are any specific parts which you had suggested but we might have missed out, or any other specific clarity issues with the new revision, we would certainly be happy to update the paper.
>
> > Modality and interaction heterogeneity measures both look very similar considering the difference of the losses
>
> They are indeed similar, but we are curious why that is seen as a weakness. We do not claim to propose fundamentally different estimators for modality and interaction heterogeneity, but rather the same underlying idea (transfer performance) applied in unimodal and bimodal settings, resulting in 2 metrics. Our motivation is to show that measuring both unimodal and interaction heterogeneity is useful by informing unimodal and crossmodal parameter sharing, which results in strong performance. We also ablated each metric individually in Table 5 as compared to full sharing and found that estimating both heterogeneity measures help.
>
> > I don't see here a convincing justification of such an approach.
>
> We have shown through our empirical results that training a heterogeneity-aware multimodal multitask model outperforms task-specific and completely shared models in the overall tradeoff between performance and efficiency. Through the examples of estimated modality and interaction heterogeneity (Figure 5), we also show natural groupings of modalities and tasks that align with human intuition - certain tasks are outliers (AV-MNIST has high difficulty transferring to others), the same modality and modality pairs across different tasks are generally similar to each other, and that there is generally more interaction heterogeneity than unimodal heterogeneity. We therefore believe that this is a natural approach, aligned with human intuition, achieves strong results, and can be sped up through low-rank approximation.
>
> > The method is combinatorial with the number of modalities, so poorly scalable.
>
> We have added experiments on this, showing the magnitude of eigenvalues after SVD on the heterogeneity matrices in Appendix A.2 - using a low-rank approximation of r = 3 is sufficient to approximate the entire matrix, so running a O(M) random sample of pairs of heterogeneity values, and imputing the rest of the matrix, is sufficient in practice. Furthermore, we do not estimate heterogeneity beyond pairs (groups of 3 or 4 modalities) which is where the real computational challenges lie, because existing multimodal models also do not go beyond unimodal and bimodal cross-attention. Finally, as compared to customary approaches of training a single shared large model (which may use more than 10 times the parameters and longer training), or individual task-specific models (in which each task can have more than 20/30 model options), our approach does not significantly increase compute time.
>
> > Assumption of distances as metrics only valid for some percentage of the time.
>
> As we have shown through experiments, the distances need not strictly be a metric for us to use them to decide parameter sharing, and that the benefits of it being a metric (i.e., use of efficient low-rank approximations due to symmetry and triangle inequality) since >86% of the time triangle inequality is satisfied (see last paragraph of Appendix A1). Knowing the exact properties of the pretrain-finetune paradigm remains a big open question in modern deep learning, which is outside the scope of the paper.
>
> > The complexity of the method is difficult to understand which its actual contributions,
>
> The main contribution is measuring modality heterogeneity to decide which unimodal parameters to share, and measuring interaction heterogeneity to decide which crossmodal parameters to share. The architecture itself is standard from a Multimodal Transformer sense, albeit with some efficient engineering and implementation (which we do not claim as our main contribution).

---

> > ### Author Response · Authors · 2023-03-24
> > **Author response to review after rebuttal - part 2**
> >
> > > Results do not seem significantly better in a statistical sense.
> >
> > Our overall results can be summarized as ‘optimizing for performance results in a model as strong as current state-of-the-art models while using 8× fewer total parameters, and optimizing for efficiency results in a model that reaches within 96% of current state-of-the-art performance but using 30× fewer total parameters’ (Figure 6 and Table 2). Note that we did not claim SOTA performance results (in which we match with close standard deviations) but only claim SOTA on the tradeoff between performance and efficiency.
> >
> > To the best of our knowledge, we are also the first to show transfer from one multimodal task to another with very different source and target modalities (Table 3). Again, we did not claim SOTA on these datasets since the standard deviations are close for some datasets, but we simply want to highlight the potential for pretraining on very different source modalities and transferring to target modalities, which is useful for low-resource target modalities.

---

### Review · Reviewer_Yccr · 2023-02-26

**Summary Of Contributions:**

This paper aims to learn a single model that can perform a wide variety of multi-modal tasks (eg a prediction task that depends on audio and video inputs). Following previous work, they first use a transformer that takes as input a sequence of (modality-specific) tokens and they then combine the outputs of that unimodal transformer using crossmodal attention. The main novelty is that they try to automatically learn which (single or pairs of) modalities should be processed using the shared weights. Specifically, the main contributions of the papers are:
- decide whether to process two modalities with shared weights, if training on one modality after training on the other helps.
- decide whether to process two pairs modalities with shared weights, if training on one pair after training on the other helps.
- show experimental gains (caveat: I do not know this benchmark )

**Audience:**

Yes

**Claims And Evidence:**

No

**Requested Changes:**

**Critical**
- **Misleading bold numbers** Your bolding in tables is not consistent inter and even intra-tables (let alone with the common approach). Table 2 you say that you bold your method's numbers if you are beating SOTA but then you should not bold UR-FUNNY and MIMIC. Furthermore such bolding is very misleading, please use the common approach of bolding the number that is statistically significantly better than others (ie don't bold the aforementioned datasets but bold SOTA for MOSEI and AV-MNIST). In table 3 you just bold your method (eg MOSEI w/o cross-modal performs better). Please Bold again statistically significantly better methods (or at least the best method). Please do so in all your tables (eg also table 12).
- **Making your metric and use of V information more precise/correct** see above.
- **Discussing how your method scales with M** see above.
- **Adding the baseline** if it is applicable.

I will update my review of "Claims And Evidence" if you at least change the first two points.

**Suggested**
- As suggested above I would actually completely drop the use of V-information given that you never use any of its properties and drop the V marginal entropy. I.e. you use a metric that could be explained in much simpler terms to the ML community without losing anything. This is a major change, which is not critical but would really improve the clarity of the paper.

**Minor**:
- page 3: "be invariant" -> "decrease"
- page 4: "metric space" -> "metric" or "distance function"

**Strengths And Weaknesses:**

# Strengths
- **Method is new**
- **Interest to the community**: building efficient multi-modal algorithms is of interest to the community. This paper shows one way of doing it by using learning what modalities to parameter tie.
- **Code is available**
- **Empirical claims are generally supported** by evidence. There are some exceptions that I highlight in "requested changes"

# Weaknesses
- **Lack of clarity and unnecessarily complicated discussion of the proposed metrics** I generally really like the concept of V-information but for this paper, it has two main issues:
    1. **It is an overly complicated repackaging of a much more common term** Indeed, given that you assume that the marginal V-entropy is constant and that you do not use any properties of V-information, all you do is look at the standard loss of a predictor (eg cross-entropy loss of a classifier). So instead of saying that you approximate a difference of V-information terms and spending one page defining that, it would be more correct/concise/simpler if you said what you actually did: "we consider how much pretraining on one modality helps training on other".
    2. **Your use of V-information is imprecise or wrong** E.g.
        - What do you mean by $\mathcal{V}(X_1)$? The family of networks is independent of the initialization and so your equation 3 is always 0, unless you condition the family on the exact optimization pipeline (algorithm/epochs/...). Is that what you meant? If that's the case then one of your properties is wrong (see below).
       - Wrong property $d(X_1;X_2) \geq 0$ with equality if $X_1=X_2$. Assuming that you define $\mathcal{V}$ as I did above then it is neither true that $d(X_1;X_2) \geq 0$ nor that $d(X_1;X_1) = 0$. Indeed, if you condition $\mathcal{V}$ on the number of steps then training from a better initialization will actually improve performance because you essentially train for longer. I.e. you will nearly always have $d(X_1;X_1) < 0$.
       - I did not think much about the triangle inequality but it likely doesn't hold for the same reason as non-negativity.
       - More generally please provide more precise definitions for your main metrics (in particular $\mathcal{V}(X_1)$) and at least provide some evidence (ideally a proof) for properties that you claim.
- **Missing discussions on computational efficiency** I might have missed it but I do not see any discussion on the computational efficiency of your metrics. If I understood correctly, for the unimodal case you compute the heterogeneity measure $d(X_1;X_2)$ for every pair of the modalities which requires training M^2 + M models (M corresponds to the unimodal models and M^2 to the unimodal models initialized from another modality). Am I missing something? if not your method scales badly with the number of modalities. Regardless you should talk about how your method scales with M (both for uni and cross-modal).
- **Missing an essential baseline** If I understood correctly that you need to train M^2 + M unimodal models (see above), then there is a more efficient baseline that I think is more natural and likely performs better. Namely to first train M unimodal models and then all M choose 2 models trained on pairs of modalities. For a total of (M^2-M)/2 models. This is more efficient and likely better given that it is closer to how the models will be used. Please clarify if I missed why such a baseline would not be applicable.

---

> ### Author Response · Authors · 2023-03-05
> **Thank you for your feedback, we have revised our paper accordingly**
>
> Thank you for your valuable feedback and insightful comments! We respond to some concerns below:
>
> > On V-information and distance metrics.
>
> We appreciate your comment. Following your suggestion, **we have changed the formulation of heterogeneity to how much pretraining on one modality helps transfer to another (see updated section 2.1)**, and removed the content on V-information. We have also clarified the scenarios when such a distance equation satisfies metric properties and provided proofs for these requirements (Appendix A.1). We believe that the newer iteration of the method is more precisely defined, and we thank your suggestions for inspiring these changes. We do need to make some assumptions to ensure that the distance is exactly a metric, and these assumptions may not hold in practice since the exact properties of the pretrain-finetune paradigm remains a big open question in modern ML. **We have included detailed discussions of this in Appendix A.1** and showed that these properties (e.g., triangle inequality) hold >90% of the time in practice, which enable low-rank approximations of the heterogeneity matrices for computational efficiency (more in the next point). Our method also works well in practice enabling positive transfer by pretraining on very different modalities and tasks to help downstream modalities and tasks.
>
> > Discussions on computational efficiency and baseline.
>
> Your suggested baseline is exactly the one we have implemented: the modality heterogeneity matrix for $M$ unique modality and task pairs has $M(M-1)/2$ unique entries after removing the upper triangular portion due to symmetry and diagonal entries since $d(X_i,X_i) = 0$. Computing these $M(M-1)/2$ entries exactly requires one to first train $M$ unimodal models (to estimate the $M$ before fine-tuning $M(M-1)$ transfer models - note that it is $M(M-1)$ rather than $M(M-1)/2$ as you pointed out since d(X_i;X_j) requires estimating both transfer directions. However, we find that a key approximation can be made in practice: the heterogeneity matrices are highly structured due to distances approximately satisfying the triangle inequality, which implies that we do not need to compute all entries and instead rely on low-rank reconstruction from partial entries in practice. In our experiments, even using a low-rank approximation of $r=3$ is sufficient to approximate the entire $10 \times 10$ matrix, so running a random sample of $O(M)$ pairs of heterogeneity values, and imputing the rest of the heterogeneity matrix, is sufficient in practice. **We have updated section 2.1 with a discussion of complexity and section 3.1 with additional details on this approximation.**
>
> > Misleading bold numbers
>
> Thank you for pointing this out - we apologize and have fixed the bolding in Tables 4 and 5 (used to be Tables 2 and 3). We have also included additional results including mean and standard deviation in the results for **Tables 2, 3, 4, and 5** and only bolded numbers that are outside 1 standard deviation of another compared model.
>
> > Minor changes
>
> Thank you for pointing these out, we have fixed them.

---

> > ### Author Response · Authors · 2023-03-30
> > **follow-up to revisions**
> >
> > Dear reviewer, thank you again for your valuable feedback and insightful comments. Please let us know if our revisions have addressed your comments regarding the presentation of results and description of information transfer (rather than v-information), as well as the additional discussion of computational efficiency. If there are any additional parts that you would like us to clarify or edit, we will be happy to look into them as well.

---

### Review · Reviewer_mTD3 · 2023-02-26

**Summary Of Contributions:**

The paper presents a novel multimodal representation learning method that quantifies multimodal and interaction heterogeneity using ¥Nu-usable information. The experiment on a multimodal dataset shows that the proposed HighMMT method can efficiently fuse multimodal data.



**Audience:**

Yes

**Broader Impact Concerns:**

Not applicable.


**Claims And Evidence:**

Yes

**Requested Changes:**

Overall, the paper is well-structured and well-written, and the study is well-motivated. However, there are a few minor issues that could be addressed.


1. "High modality
I am not sure that "high-modality" is an appropriate term to describe a situation where a robot has a large number of sensor modalities. I have never seen the term before. To be honest, the term gives me a very different understanding of the existence of a large number of modalities. If it is not a keyword accepted by the community, it will be better to rephrase the term.

2. Realistic robotics scenario
The author often mentions the scenario of "robotics".
In robotics, the target task is not only label estimation, but also control and so on. Also, important heterogeneity is the temporal gap of data, data sparsity, unpredictable missing data, and so on. A robot needs to update its knowledge by interacting with the environment.

I think the paper is also an important contribution to the robotics community. However, there is still a big gap between multimodal information fusion in real robots and the task tested in the paper. Therefore, if the authors provide some more concrete ideas to bridge the gap between machine learning and the robotics community, it will attract more robotics researchers to read and cite this paper.

Also, I think this research is very relevant to world models and predictive coding in robotics, which emphasizes sensorimotor interaction with the robotic world. If possible, please discuss the applicability of the proposed method to world models.

World models and predictive coding for cognitive and developmental robotics: Frontiers and Challenges
https://arxiv.org/abs/2301.05832

3. Systematic review of Multimodal representation learning

As the authors mention, there are many ways to integrate multimodal information. It might be nice for readers to be given a systematic overview of multimodal fusion or explicit references to review papers. The following might be a good candidate for this.


Suzuki, Masahiro and Yutaka Matsuo. "A review of multimodal deep generative models". Advanced Robotics 36.5-6 (2022): 261-278.
https://www.tandfonline.com/doi/abs/10.1080/01691864.2022.2035253

**Strengths And Weaknesses:**

+ The paper provides a new method that improves the fusion of multimodal information based on solid theoretical background.
+ The proposed method is evaluated using a multimodal dataset and showed significant improvement.

- The performance is only evaluated on classification tasks on a static dataset.
- The applicability to the real data in robotics has not been explored.

---

> ### Author Response · Authors · 2023-03-05
> **Thank you for your feedback, we have revised our paper accordingly**
>
> Thank you for your valuable feedback and insightful comments! We respond to some concerns below:
>
> > The term ‘high-modality’
>
> We believe that the new term 'high-modality' can help us to differentiate the large and diverse set of modalities that we would like to learn representations over (in contrast to current work studying a fixed and small set of modalities like images, text, and audio), which introduces new challenges of complexity since adding new model parameters for every new modality or task becomes expensive, as well as representation learning over diverse and partially-observable modality inputs. Our approach of heterogeneity quantification and modeling is a step in this direction to guide parameter sharing and representation fusion in high-modal scenarios. To the best of our knowledge, there is no current term emphasizing the technical challenges revolving over many diverse modality inputs. We have **expanded on our definitions of high-modality in the introduction and section 2 (discussion of computational complexity)** to make them more precise, in order to prevent any confusion around this new term.
>
> > Realistic robotics scenarios and motivations
>
> Thank you for your suggestion. We have added a **new paragraph in Appendix C.1** highlighting your suggested references on multimodal learning in robotics. Our paper also includes initial results on robotics tasks using the Vision & Touch [1] and MuJoCo Push [2] datasets which record the manipulation of simulated and real robotic arms equipped with visual (RGB and depth), force, and proprioception sensors. In MuJoCo Push, the goal is to predict the pose of the object being pushed by the robot end-effector. In Vision & Touch, the goal is to predict action-conditional learning objectives that capture forward dynamics of the different modalities (contact prediction and robot end-effector pose). Both of these datasets include visual, sensor, and contact data collected from real robotics and published at top robotics conferences. Multisensor fusion in robotics is especially important due to the large number of sensors, requirements for robust learning from noisy sensors, and limited data as compared to tasks involving images, text, and audio. From Table 4, our multimodal multitask approach does indeed perform well on these initial robotics tasks, and we do hope to deploy some of these models on real-world robotic control tasks as future work.
>
> [1] Michelle A Lee, Yuke Zhu, Krishnan Srinivasan, Parth Shah, Silvio Savarese, Li Fei-Fei, Animesh Garg, and Jeannette Bohg. Making sense of vision and touch: Self-supervised learning of multimodal representations for contact-rich tasks. ICRA 2019, IEEE Transactions on Robotics 2020.
> [2] Michelle A Lee, Brent Yi, Roberto Martín-Martín, Silvio Savarese, and Jeannette Bohg. Multimodal sensor fusion with
> differentiable filters. IROS 2020.
>
> > References on world models, predictive coding, and multimodal generative models.
>
> Thank you for these references, we have added these to our **related work (section 4)**. You are right that the related work in multisensor fusion for robotics, healthcare, self-driving cars and so on are indeed relevant to our approach. HighMMT also shows strong initial results on some of these domains (robotics and healthcare).

---

### Author Response · Authors · 2023-03-05
**Thank you to all reviewers for your valuable feedback! Summary of revisions made to submission**

Dear reviewers, we are extremely grateful for your valuable feedback and insightful comments. Your concrete suggestions are a valuable step in this direction, and we have revised our submission accordingly to take these into account. In this short note we summarize the main changes to the latest revision of our submission (the main changes are on pages 2,3,4,5,6 concerning improved description of the method and additional experiments in Tables 2,3,4,5). We have also included the updated Appendix with additional results and details. All updates are highlighted in red.

1. Reviewer mTD3: We have added the suggested references and discussions regarding multisensor fusion in robotics.
2. Reviewer Yccr and tic5: Following your suggestion, we have changed the formulation of heterogeneity to how much pretraining on one modality helps transfer to another **(section 2.1)**, and removed the content on V-information. We have also clarified the scenarios when such a distance equation satisfies metric properties **(Appendix A.1)**.
3. Reviewer Yccr: We added discussion of computational complexity in **section 2.1**, and further discussed strategies for approximating the heterogeneity matrix using low-rank approximations in **Appendix A.2** for efficiency.
4. Reviewer tic5: We have updated **section 2.2** to include relevant equations and figures showing the model architecture and workflow, moving a lot of critical content from the Appendix to the main paper.
5. Reviewer tic5: We added additional results that carefully ablate the number of clusters k on performance in **Table 2.**
6. Reviewer Yccr and tic5: We included additional results including mean and standard deviation in the results for **Tables 2, 3, 4, and 5**, and fixed the bolding issues in Tables 4 and 5.

---

### Author Response · Authors · 2023-05-17
**Thank you for the extremely detailed and valuable feedback - Camera-ready revision**

Dear Action Editor and Reviewers,

Thank you so much for your detailed reviews and constructive feedback. We have uploaded the camera-ready version with the following changes:
- Fixed Figure 3 and Equations 9 and 10 to clarify that Z is the result of recursive updates via cross-attention and self-attention across multiple layers, using $Z^{(0)}$ as initialization and $Z^{(L-1)}$ to $Z^{(L)}$ as the result of cross-attention and self-attention at the $L$th Perceiver layer.
- Fixed Table 3 to include few-shot description, and edited the corresponding paragraph to clarify that the selection of datasets to pre-train and transfer is from a fixed order ranked by largest dataset size (number of datapoints x memory size per datapoint) with reference to Appendix D.1 for details.
- Included in Table 4 and 5 that means and standard deviations in tables are based on 10 runs, and also added this to the main text when describing Tables 2 and 3.
- Updated Figure 10 caption in Appendix for time-aligned and fixed the typo.
- Added link to our public code repository with the datasets and models from our experiments.

The interactions on TMLR have been a great and positive experience - thank you again and best wishes!

---

### Decision · Action_Editors · 2023-04-12

**Recommendation:** Accept with minor revision

**Comment:**

This paper underwent a significant revision based on the comments from reviewer Yccr, who proposed that the authors discard the original formulation of the method based on $\mathcal{V}$-information, and reviewer tic5, who called out many issues with the organization of the paper (too much important information relegated to appendices), clarity (better motivation for training procedure, better explanation of ablations), more substantiation of the approximate metric properties of the heterogeneity measures, and clearer relation to SOTA results.

Following the revision and discussion, reviewer tic5 still had the same concerns, while the other two reviewers were satisfied. I therefore carefully read the revised paper myself and went point-by-point through the issues raised by reviewer tic5. In my judgement, the revised paper does adequately address the concerns raised. While I am not following reviewer tic5's final recommendation, I thank this reviewer for the careful and thorough review that helped to greatly improve the quality of the paper, and to reviewer Yccr for recommending a reframing of the heterogeneity measures without recourse to $\mathcal{V}$-information.

However, I did find a small number of issues listed below that should be addressed in the final version of the paper.

- Figure 3 and Equations 9 and 10: it is not ideal to use "Z" for both the latent vectors used by the perceiver and the output of the unimodal encoder without some notation to indicate that Z is the result of an unrolled recurrence.
- Table 3: It would be helpful to remind the reader that these are few-shot transfer results (as stated in the reply to Reviewer tic5). Also, in the case of pretraining on one task, there are three tasks to choose from, and likewise when pretraining on two tasks there are three task pairs to choose from. What choices were made for the results in Table 3?
- The paper should explain somewhere that means and standard deviations in tables are based on 10 runs.
- In the caption to Fig. 11 (in Appendix), remind reader that the modalities are time-aligned. Also fix typo: "common attention patters" -> "common attention patterns"


**Audience:**

Models that can efficiently learn from a broad range of modalities and that allow for good tradeoffs between model performance and cost will be of interest to at least some portion of the TMLR audience.


**Claims And Evidence:**

The main claims of the paper are listed below, along with evidence provided in the paper to support the claims.

**Claim**: The disparity between two modalities $X_1$ and $X_2$ may be measured by the difference in loss between two models trained on some task with targets $Y$: a test model in which parameters are initialized from a model trained with samples $(X_1; Y)$ and a baseline model trained from randomly initialized parameters and samples $(X_2; Y)$. Denoting this disparity $T(X_{1} \rightarrow X_{2};Y)$, define a heterogeneity measure $d(X_{1}; X_{2}) = max(T(X_{1} \rightarrow X_{2};Y), 0) + max(T(X_{2} \rightarrow X_{1};Y), 0)$. $d(X_{1}; X_{2})$ is approximately a metric in that it is usually nonnegative, symmetric by construction, and usually satisfies a relaxed version of the triangle inequality. The heterogeneity of pairs of modalities $(X_1, X_2)$ and $(X_3, X_4)$ (called "interaction heterogeneity") may be characterized via a similar procedure.

**Evidence**: Discussion in Section 2.1, Figure 5, and the discussion in Appendix A.1.

**Claim**: The approximate metric characteristics of $d(X_{i}; X_{j})$ and $d(X_{i}, X_{j}; X_{k}, X_{\ell})$ lead to the modality heterogeneity matrix and interaction heterogeneity tensor for a suite of $M$ modality/task pairs having low-rank structure, which in turn licenses an efficient $O(M)$ estimation procedure for the modality and interaction heterogeneities.

**Evidence**: Discussion in Section 2.1, Figure 5, Appendix A.2, and Figure 7 (in Appendix).

**Claim**: These heterogeneity measures can be used to specify parameter sharing in a multimodal transformer model based on the Perceiver, the parameter sharing allows for a tradeoff between model size and performance, and the tradeoffs achieved via the heterogeneity measurements are better than alternative approaches.

**Evidence**: Figure 6, Table 2, and Table 5.

**Claim**: The proposed modeling approach achieves positive transfer to new modalities and tasks.

**Evidence**: Table 3

**Claim**: The proposed modeling approach is competitive (in some cases) with SOTA task-specific models.

**Evidence**: Table 4

In my judgement, claims in the paper are adequately supported.